# HELM: Hierarchical Encoding for mRNA Language Modeling

**Mehdi Yazdani-Jahromi** [*][†]
University of Central Florida
yazdani@ucf.edu

**Mangal Prakash** [*]
Johnson & Johnson Innovative Medicine
mpraka12@its.jnj.com

**Tommaso Mansi**
Johnson & Johnson Innovative Medicine
tmansi@its.jnj.com

**Artem Moskalev** [‡]
Johnson & Johnson Innovative Medicine
amoskal2@its.jnj.com

**Rui Liao** [‡]
Johnson & Johnson Innovative Medicine
rliao2@its.jnj.com

## ABSTRACT

Messenger RNA (mRNA) plays a crucial role in protein synthesis, with its codon structure directly impacting biological properties. While Language Models (LMs) have shown promise in analyzing biological sequences, existing approaches fail to account for the hierarchical nature of mRNA's codon structure. We introduce Hierarchical Encoding for mRNA Language Modeling (HELM), a novel pre-training strategy that incorporates codon-level hierarchical structure into language model training. HELM modulates the loss function based on codon synonymity, aligning the model's learning process with the biological reality of mRNA sequences. We evaluate HELM on diverse mRNA datasets and tasks, demonstrating that HELM outperforms standard language model pre-training as well as existing foundation model baselines on seven diverse downstream property prediction tasks and an antibody region annotation tasks on average by around 8%. Additionally, HELM enhances the generative capabilities of language model, producing diverse mRNA sequences that better align with the underlying true data distribution compared to non-hierarchical baselines.

## 1 INTRODUCTION

RNA analysis is becoming increasingly important in molecular biology (Liu et al., 2023; Fu, 2014). Messenger RNA (mRNA) is of particular interest due to its unique role in protein synthesis (Sahin et al., 2014). mRNA consists of triplets of nucleotides, called codons, which directly translate to protein amino acids through a surjective many-to-one mapping (Figure 1 *left*). This results in multiple synonymous mRNAs that encode identical amino acid sequences while exhibiting distinct physical and biological properties at the nucleotide level (Buhr et al., 2016). The role of mRNA, serving as an intermediary between DNA and proteins, underlies its therapeutic potential in applications such as vaccines and gene therapies (Caplen, 2004; Pardi et al., 2018), while also presenting challenges for analysis and optimization (Liu et al., 2021; Xu et al., 2025).

Language Models (LMs) have emerged as powerful tools for analyzing biological sequences, with notable successes in protein (Elnaggar et al., 2021; Ferruz et al., 2022; Lin et al., 2023; Hie et al., 2024) and DNA (Nguyen et al., 2024a; Zhou et al., 2023) research. Despite the importance of mRNA, the field still lacks specialized LMs tailored for its analysis. Existing RNA LMs (Li et al., 2023; Chen et al., 2023) focus on non-coding sequences and do not account properly for codon

---

[*]Equal contribution as first authors.
[†]This work was done while the author was an intern at Johnson & Johnson.
[‡]Equal contribution as last authors.

hierarchy (Fig. 1 *right*) which, as we demonstrate, falls short when dealing with mRNA tasks. In this work, we aim to address this gap in mRNA language modeling by focusing specifically on the unique challenges presented by mRNA sequences.

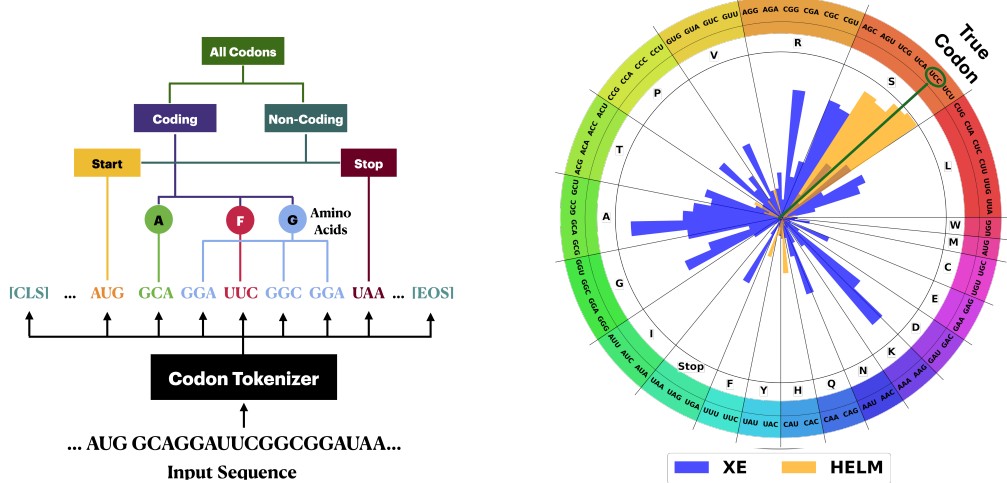

Figure 1: Hierarchical encoding of mRNA sequences as a biological prior. ***Left:*** Hierarchical structure of codons for HELM and codon tokenization. The tree diagram illustrates the codon hierarchy used in the HELM approach, categorizing codons into Start, Coding (grouped by amino acids), and Stop. This hierarchy informs the loss calculation. The codon tokenizer demonstrates the process of converting an mRNA input sequence into codon tokens for modeling. ***Right:*** Codon prediction probabilities on a amino acid codon wheel. Segments represent amino acids, bars represent codons. Orange: HELM approach; Blue: cross-entropy (XE) loss. Bar height indicates probability. Non-hierarchical XE model assigns high probabilities to non-synonymous codons for masked tokens, while HELM assigns high probabilities to synonymous codons, even when errors occur.

To address the limitations of existing bio-language modeling methods, we introduce Hierarchical Encoding for mRNA Language Modeling (HELM), a novel pre-training strategy for mRNA sequences. Rather than directly adopting natural-language pre-training methods such as Masked Language Modeling (MLM) (Devlin et al., 2019; Liu et al., 2020c; Cheng et al., 2024) or Causal Language Modeling (CLM) (Radford, 2018; Radford et al., 2019; Brown, 2020), HELM begins by recognizing and formalizing the hierarchical structure of mRNA sequences based on codon synonymy. HELM modulates the CLM or MLM loss based on a codon's position in this hierarchy, treating errors between synonymous codons as less significant than those resulting in different amino acids. This approach requires minimal modifications to standard pipelines while aligning the model training process with the intrinsic hierarchical structure of mRNA data (Figure 1 *right*). In addition, to facilitate further development of mRNA LMs, we provide a consistent comparison of various tokenization methods, hierarchical and standard pre-training strategies, and different language model architectures, including recent Mamba and Hyena models (Gu & Dao, 2023; Poli et al., 2023).

To demonstrate the practical advantages of HELM, we focus on the important domain of antibody mRNAs where codon distribution can significantly impact physical and therapeutic properties (Zhang et al., 2023). We conduct comprehensive experiments, evaluating our hierarchical model against its non-hierarchical counterparts pre-trained with natural-language CLM and MLM as well as against other state-of-the-art RNA foundation models. Our results show that hierarchical pre-training yields significantly better representations of antibody-encoding mRNA and even excels on non-antibody mRNA data. Our hierarchy-aware model outperforms non-hierarchical counterparts pre-trained with natural-language CLM and MLM as well as other state-of-the-art RNA foundation models by around 8% on average across seven diverse downstream property prediction tasks and an antibody region annotation tasks while requiring up to 2 times fewer model parameters. Beyond its excellent discriminative performance, we demonstrate that HELM-based CLM pre-training yields strong generative models capable of producing diverse mRNA sequences while preserving closer alignment with the true data distribution. Ultimately, HELM addresses the limitations of current bio-LMs by effectively capturing the hierarchical structure of mRNA as a strong biological prior,

leading to improved performance in both predictive and generative tasks while requiring minimal modification of the pre-training.

To sum up, we make the following contributions:

- We highlight the importance of biological hierarchy when modeling mRNA sequences and demonstrate that standard natural-language pre-training methods do not account for this inherent biological structure, leading to suboptimal performance in mRNA tasks.

- We propose Hierarchical Encoding for mRNA Language Modeling (HELM), a novel pre-training strategy that incorporates mRNA hierarchical structure into language model pre-training. HELM modulates the CLM or MLM loss based on codon hierarchy, aligning the model's learning process with the biological structure of mRNA sequences.

- We provide a consistent comparison of various tokenization methods, hierarchical and standard pre-training strategies, and different LM architectures to facilitate further development of mRNA models. We also introduce a large high-quality curated mRNA dataset to help further research for mRNA language modeling.

- We demonstrate the advantages of representation learning with HELM where our method yields substantial downstream performance improvement on multiple diverse mRNA datasets spanning property prediction, automated sequence annotation and sequence generation tasks.

## 2    RELATED WORKS

**Pre-training bio-language models**    Recent advancements in LMs have significantly improved the analysis of biological sequences such as DNA, RNA, and proteins (Laurent et al., 2024; Prakash et al., 2024). The standard approach for bio-language modeling adopts pre-training strategies from natural language; state-of-the-art protein or DNA LMs rely on Masked Language Modeling (Liu et al., 2020c; Lin et al., 2023; Dalla-Torre et al., 2023) or Causal Language Modeling (Brown, 2020; Lin et al., 2023) and pre-train representations on large bio-databases (Bairoch et al., 2005; Suzek et al., 2007; Consortium et al., 2013; Schoch et al., 2020). In addition, the modeling choices of pre-training bio-LMs involve network architecture and tokenization strategy. Regarding architectures, transformer-based models have been widely adopted (Elnaggar et al., 2021; Lin et al., 2023), while recent works explore structured state-space (Peng et al., 2024; Sgarbossa et al., 2024) and long-convolutional models (Nguyen et al., 2024b;a; Moskalev et al., 2024) to better handle long-range dependencies in biological sequences. Tokenization strategies vary from character-level approaches (Elnaggar et al., 2021; Lin et al., 2023) to k-mer tokenization (Dalla-Torre et al., 2023) and Byte Pair Encoding (Zhou et al., 2023). These existing pre-training and modeling choices primarily adapt natural language paradigms to biological sequences, overlooking the incorporation of inherent biological structures crucial for mRNA analysis. Moreover, the lack of consistent comparisons between these various approaches hinders the development of effective mRNA LMs. Our work addresses these limitations with HELM, a pre-training strategy incorporating mRNA's hierarchy into the training, and we also provide analysis of various pre-training and modeling choices to facilitate further development of mRNA LMs.

**RNA language models**    Unlike the many protein and DNA LMs (Lin et al., 2023; Elnaggar et al., 2021; Dalla-Torre et al., 2023; Nguyen et al., 2024a), there is a noticeable lack of models specifically designed for mRNA due to the smaller, specialized datasets available. Existing RNA models focus on specific RNA types or regions, such as RNA-FM (Chen et al., 2022) and RINALMO (Penić et al., 2024) for non-coding RNAs, UTR-LM (Chu et al., 2024) and UTRBERT (Yang et al., 2023) for untranslated regions, and SpliceBERT (Chen et al., 2023) for precursor mRNA. While these models are valuable, they are not tailored for protein-coding mRNAs. Proprietary models such as CodonBERT (Li et al., 2023) have been developed but they do not utilize the natural hierarchy of mRNA sequences. Our work addresses these limitations by introducing HELM to explicitly incorporate the biological knowledge of mRNA structure into the pre-training process of a LM.

**Modeling hierarchical data**    Hierarchical relationships are important across various domains, and several methods have been proposed to model them effectively. In computer vision, approaches

range from embedding-based methods like DeViSE (Frome et al., 2013), which maximizes cosine similarity between image and label embeddings, to geometric approaches that rely on hyperbolic spaces (Liu et al., 2020b; Atigh et al., 2022) or hyperspheres with structural constraints (Barz & Denzler, 2019; Bengio et al., 2010). Hierarchical loss functions, such as hierarchical cross-entropy (HXE) (Bertinetto et al., 2020; Garg et al., 2022), have been applied to tasks like image classification and astrophysical transient classification (Villar et al., 2023). In natural language processing, hierarchical structures have been modeled both implicitly through contrastive learning (Liu et al., 2020a; Gosselin & Zouaq, 2023) and explicitly using hyperbolic losses (He et al., 2024) or hierarchical softmax (Schuurmans & Frasincar, 2023). While these methods show great promise in their respective domains, they are not tuned for biological data. Our HELM approach addresses this gap by extending the hierarchical learning to mRNA sequences, effectively capturing the codon hierarchy in the pre-training process. This demonstrates the potential of incorporating biological priors into language modeling pre-training, paving the way for more accurate and biologically relevant models in this domain.

## 3 HIERARCHICAL CROSS-ENTROPY LOSS

Hierarchical cross-entropy (HXE) is a technique for training neural networks that incorporates hierarchical information into the learning process. HXE was originally proposed in computer vision for image classification (Bertinetto et al., 2020) tasks, where the data labels can be organized in a hierarchical structure, such as a tree. When the hierarchy $H$ is structured as a tree, HXE allows for a unique factorization of the categorical distribution $p(C)$ over the classes in terms of the conditional probabilities along the path from each class to the root of the tree. Specifically, for any leaf node/class $C$ with a path $C^{(0)} = C, \ldots, C^{(h)} = R$, the probability of $C$ can be written as $p(C) = \prod_{l=0}^{h-1} p(C^{(l)} \mid C^{(l+1)})$, where $h$ is the height of the node $C$. The conditional probabilities can be expressed in terms of the class probabilities as:

$$p(C^{(l)} \mid C^{(l+1)}) = \frac{\sum_{A \in \text{Leaves}(C^{(l)})} p(A)}{\sum_{B \in \text{Leaves}(C^{(l+1)})} p(B)}, \tag{1}$$

where $\text{Leaves}(C)$ denotes the set of leaf nodes within the subtree rooted at $C$. To incorporate hierarchical information directly into the loss function, the classifier's output can be factorized according to the hierarchical structure, and the total loss can be defined as a reweighted sum of the cross-entropies of these conditional probabilities along the hierarchical path. This leads to the hierarchical cross-entropy (HXE) loss, effectively incorporating the hierarchical information into the learning process:

$$L_{HXE}(p, C) = -\sum_{l=0}^{h-1} \lambda(C^{(l)}) \log p(C^{(l)} \mid C^{(l+1)}), \tag{2}$$

where $\lambda(C^{(l)})$ is the weight associated with the edge between nodes $C^{(l+1)}$ and $C^{(l)}$.

**Formalizing the hierarchcy for mRNA sequences** We present a novel formalism for constructing a multi-level hierarchical tree $H = (V, E)$ that captures the structured relationships within mRNA sequences at the codon level (see Fig. 1 left, Appendix Fig. 6) and well-rooted in biology (Clancy & Brown, 2008). The root node $R$ of this tree represents a node corresponding to all codons. This root node has two immediate child nodes: coding codons $C_{\text{coding}}$ and non-coding codons $C_{\text{non-coding}}$. The non-coding codons $C_{\text{non-coding}}$ further branch into two child nodes: the start codon node $C_{\text{start}}$ and the stop codon node $C_{\text{stop}}$. The start codon node $C_{\text{start}}$ has a single leaf node corresponding to the codon AUG. The stop codon node $C_{\text{stop}}$ has three leaf nodes corresponding to the codons UAA, UAG, and UGA. On the other side of the hierarchy, the coding codons $C_{\text{coding}}$ are organized into multiple child nodes, each corresponding to a specific amino acid $A_j$. Each amino acid node $A_j$ has a number of child nodes, each corresponding to a synonymous codon that encodes for $A_j$. Let $\text{Codons}(A_j) = \{C_{i_1}, C_{i_2}, \ldots, C_{i_m}\}$ denote the set of all synonymous codons for the amino acid $A_j$. These synonymous codons preserve the same amino acid, allowing for degeneracy in the genetic code. The complete hierarchical structure is thus represented by the tree $H = (V, E)$, where $V$ includes the root node, all internal nodes (corresponding to coding and non-coding codons, start/stop codons, and amino acids), and all leaf nodes (corresponding to individual codons). The edges $E$ define the hierarchical relationships between these nodes.

**Hierarchical MLM and CLM pre-training for mRNA**    We adapt the Hierarchical Cross-Entropy to incorporate the biological structure of mRNA into Masked Language Modeling and Causal Language Modeling pre-training strategies. For the hierarchical MLM, we randomly mask a portion of codons in the input sequence and train a model to predict the masked codons supervising it by the HXE loss. In the hierarchical CLM, a model predicts the next codon given the previous ones with the HXE loss function. The hierarchical structure of mRNA informs the HXE objective, allowing the model's output token probabilities to be factorized according to the tree structure. Then, the HXE loss uses Eq.1 and Eq.2 weighting the MLM or CLM errors differently based on the codon's position in the hierarchy. We employ a practical weighting function $\lambda(C) = \exp(-\alpha h(C))$ where $h(C)$ is the height of the node $C$ in the hierarchy and $(\alpha > 0)$. The value of $\alpha$ determines how much weight is given to information at different hierarchical levels. This weighting function is applied in both MLM and CLM pre-training, allowing HELM to capture both local codon-level information and the global hierarchical structure of the genetic code. By incorporating this hierarchical loss into pre-training, HELM encourages the model to learn biologically meaningful representations of mRNA sequences, which closely aligns the model's learning with the biological structure of mRNA sequences. Note that our approach does not require modifying the architecture or tokenization of a model and permits operating on the standard codon vocabulary. mRNA hierarchy modeling with HXE will be referred as HELM from here on.

## 4    EXPERIMENTS

In our experiments, we first compare various mRNA language modeling configurations by exploring different tokenization strategies and model architectures (Transformer, Mamba, Hyena) with MLM and CLM objectives. We demonstrate how HELM significantly improves mRNA property prediction over its non-hierarchical counterparts. Next, we investigate the effectiveness of hierarchical mRNA encoding, evaluating it through synonymous sequence clustering. Following this, we present generative evaluations include assessing generated sequence quality and diversity via Frechet Biological Distance (FBD), as well as testing the preservation of functional properties in generated mRNA sequences. Finally, we conclude with antibody region annotation, showcasing the model's practical applicability in bioinformatics.

**Datasets**    Unlike protein and DNA modality which boast many large-scale datasets, mRNA does not have many curated high-quality pre-training datasets. For this reason, we curated the OAS database (Olsen et al., 2022) which contains antibody mRNA data from over 80 different studies with around 2 billion unpaired and 1.5 million paired sequences from various species. Although prior studies have curated this database on protein level (Ruffolo et al., 2021; Shuai et al., 2023; Kenlay et al., 2024) in the context of antibody-protein language modeling, a high-quality curated version of corresponding mRNA data does not exist. Due to its origin in high-throughput sequencing studies, the mRNA sequences in OAS exhibit varying levels of sequencing errors and noise. We conducted an extensive curation process to ensure high-quality mRNA data for pre-training. Dataset curation details can be found in Appendix A.14. Post-curation, we end up with 15.3 million mRNA sequences for pre-training mRNA LMs.

For downstream evaluation, we use the following seven mRNA datasets, including antibody (Ab) encoding mRNA and general mRNA sequences:

- Ab1 (Anonymous, 2024) includes 1,200 Ab-mRNA sequences with protein expression labels, which are critical for optimizing mRNA design to ensure efficient protein translation, a key factor in the development and manufacturing of mRNA vaccines, including cancer immunotherapies.

- Ab2 (Anonymous, 2024) contains 3442 Ab-mRNA sequences with protein expression labels, acquired using a different experimental platform than Ab1.

- MLOS (Li et al., 2023) has 167 viral mRNA sequences with expression in HeLa cells.

- iCodon (Diez et al., 2022) includes 65357 mRNA sequences with thermostability profiles from humans, mice, frogs, and fish.

- Tc-Riboswitches (Groher et al., 2018) consists of 355 riboswitch mRNA sequences with switching factor measurements.

- mRFP (Nieuwkoop et al., 2023) has 1459 mRNA sequences with protein production levels for various gene variants in E. coli.
- COVID-19 Vaccine (Wayment-Steele et al., 2022) has 2400 mRNA sequences with degradation labels relevant for mRNA vaccine formulations.

All downstream evaluation datasets provide regression labels for evaluating the quality of mRNA property prediction.

**Evaluation**   For iCodon, Tc-Riboswitches, mRFP and COVID-19 Vaccine datasets, we use predefined splits from prior publications to ensure fair comparison. For other datasets, we apply clustering-based train/validation/test splitting (LinClust (Steinegger & Söding, 2018) similarity threshold 0.9) to prevent data leakage. We use a train/validation/test split ratio of 70:15:15. When evaluating property prediction tasks, we use Spearman rank correlation as a standard metric under commonly used probing methodology (Marquet et al., 2022; Chen et al., 2024; Outeiral & Deane, 2024; Harmalkar et al., 2023) with a TextCNN (Kim, 2014) head to assess representation quality and transferability. For generative evaluation, we take the metrics used to evaluate generative models for DNA (Stark et al., 2024; Shuai et al., 2023) and we adopt them for mRNA data. For sequence region annotation, we report region annotation accuracy. We provide comprehensive experimental details in Appendix A.3.

## 4.1   Pre-training and modeling choices

We first aim to establish a strong non-hierarchical mRNA LM as a baseline. For this, we aim for consistent comparison between various standard pre-training strategies, different architectures, and various tokenization methods. We conduct the comparison on the property prediction mRNA tasks where we pre-train all our language language models on the same curated OAS data and test them on downstream property prediction tasks.

### 4.1.1   Experimental details

**Pre-training objective**   To train an mRNA LM, we start with two standard pre-training objectives: Masked Language Modeling (MLM) and Causal Language Modeling (CLM). MLM is an inherently bidirectional pre-training method that relies on model learning to capture context from both directions of a sequence. CLM, on the other hand, is unidirectional, predicting each token based on previous tokens. We evaluate both objectives to determine their effectiveness in capturing the biological structure of mRNA sequences.

**Model architectures**   We evaluate various state-of-the-art architectures for sequence modeling: Transformer (Radford et al., 2019), Hyena (Poli et al., 2023), and Mamba (Gu & Dao, 2023). Transformers support both MLM and CLM objectives, while Hyena and Mamba are primary designed for CLM. All models use 50M parameters, balancing performance and efficiency. We found that models of this scale can outperforms larger existing models while maintaining reasonable run-times (see Appendix A.8). Detailed pre-training information is available in Appendix A.2.

**Tokenization**   Inspired by protein and DNA language modeling, we explored various tokenization strategies for mRNA sequences: nucleotide, codon-level, and 6-mer. We observed that codon-level tokenization consistently outperforms other strategies (results in Appendix Sec. A.4). We hypothesize that the superior performance of codon tokenization stems from codons' role as the fundamental units of the genetic code, directly mapping to amino acids during protein synthesis. In addition, the codon level tokenization allows to naturally capture the variability in wobbling nucleotides, where changes in the third nucleotide do not always alter the amino acid produced (Barricelli, 1977; Outeiral & Deane, 2024). Here on, we adopt codon-level tokenization as the default strategy for our models. This way, our vocabulary includes $4^3 = 64$ codon tokens for all possible nucleotide combinations, plus special tokens for start, stop, padding, masking, and unknown characters.

**Baselines**   We additionally compare trained models against three state-of-the-art public foundation models: RNA-FM, SpliceBERT, and CodonBERT. While RNA-FM and CodonBERT are 100M parameter models, SpliceBERT is a 20M parameter model. We choose these baselines as CodonBERT is the only existing mRNA LM publicly available while SpliceBERT is pre-trained on pre-mRNA, and RNA-FM is known to be one of the strongest baselines in recent works because of its

Table 1: Performance of mRNA LMs trained with standard cross-entropy loss on downstream tasks. Our models outperform general-purpose RNA baselines across various mRNA tasks. MLM and CLM show comparable results. Bold: best performance. Italics: second-best. Missing values: model unable to process due to sequence length limitations.

| Model | Ab1 | Ab2 | MLOS | iCodon | Tc-Riboswitches | mRFP | COV-19 Vaccine |
|---|---|---|---|---|---|---|---|
| *Baseline Models* | | | | | | | |
| One-hot | 0.431 | 0.421 | 0.462 | 0.152 | 0.378 | 0.511 | 0.550 |
| RNA-FM | 0.595 | 0.515 | - | - | 0.504 | 0.527 | 0.742 |
| SpliceBERT | 0.652 | 0.542 | - | - | 0.418 | 0.596 | 0.757 |
| CodonBERT | 0.686 | 0.557 | 0.543 | 0.350 | 0.502 | 0.770 | 0.780 |
| *mRNA LMs (Ours)* | | | | | | | |
| Transformer XE (MLM) | *0.748* | **0.599** | **0.653** | 0.503 | **0.569** | 0.753 | **0.801** |
| Transformer XE (CLM) | **0.752** | *0.597* | 0.611 | 0.498 | *0.531* | 0.815 | 0.787 |
| Hyena XE | 0.743 | 0.594 | *0.623* | **0.526** | 0.517 | **0.844** | *0.797* |
| Mamba XE | 0.742 | 0.585 | 0.620 | *0.509* | 0.520 | *0.823* | 0.741 |

reported strong performance across multiple domains (Nguyen et al., 2024a; Franke et al., 2024; Boyd et al., 2023). Additionally, we include a simple one-hot encoding-based baseline to evaluate how much LMs performances surpass basic sequence representation methods commonly used (Harmalkar et al., 2023; Boyd et al., 2023).

### 4.1.2 RESULTS

We first compare the performance of the transformer-based mRNA language models to state-space Mamba and long-convolutional Hyena-based architectures. In Table 1, we can see that transformer-based models outperform Hyena and Mamba on 5 out of 7 datasets, with the transformer CLM trailing closely behind Mamba on the remaining 2 datasets. The superior performance of transformers is likely due to the relatively short sequence lengths ($\sim 1000$ tokens) of mRNA sequences in both pre-training and downstream datasets. Transformers, with their explicit global attention mechanism between each pair of tokens, excel at capturing fine-grained interactions in these relatively short sequences. In contrast, Hyena and Mamba architectures are optimized for handling much longer sequences via efficient convolutional or state-space mechanisms and may not fully leverage their strengths in this shorter sequence regime owing to their implicit handling of sequence dependencies, despite performing competitively. Given that transformer-based models generally show better performance than Hyena and Mamba models, we select transformer-based MLM and CLM models for all subsequent experiments.

Additionally, our results indicate that for the transformer models, both MLM and CLM perform comparably on 3 out of 6 datasets (Ab1, Ab2, and iCodon), while MLM outperforms CLM on MLOS and Tc-Riboswitches data, and CLM surpasses MLM on the mRFP dataset. The comparable performance of MLM and CLM across most cases suggests that the statistical and structural properties of mRNA are adequately captured by both unidirectional and bidirectional context modeling. This implies that the underlying biological information in mRNA sequences may be robust to the specific directional biases introduced by MLM or CLM pre-training, with neither approach consistently outperforming the other across varied bio-sequence contexts.

We then compare the performance of our transformer, Hyena, and Mamba models against various foundation model baselines. Table 1 illustrates that our transformer, Mamba, and Hyena-based LMs outperform all baselines by 5-17 percentage points across all downstream tasks which highlights the value of our curated mRNA pre-training dataset, tokenization, and architectural design choices.

### 4.2 HELM IMPROVES MRNA PROPERTY PREDICTION

Next, we evaluate the effect of learning hierarchical mRNA encoding with HELM for the mRNA property prediction tasks. As can be observed from Table 2, training with biological hierarchical prior consistently improves model performance for both MLM and CLM objectives on all six datasets compared to baseline models (denoted as XE) trained with standard non-hierarchical cross-entropy. Note that both XE and HELM have the same architecture, tokenization and pre-training

dataset, thus the superior performance of HELM serves as an ablation showcasing the benefit of learning hierarchy. For causal language modeling, HELM-CLM outperforms the non-hierarchical XE-CLM model in five out of six datasets. For the masked language modeling, HELM-MLM outperforms the non-hierarchical XE-MLM model in all datasets. Within hierarchy-aware models, we again observe the same trend that MLM and CLM HELM models perform comparably for most tasks with each performing best on three of the six datasets. Results in terms of Pearson correlation and $R^2$ are reported in Appendix A.11 and show similar trends. Additional analysis in Appendix Sec. A.11, conducted on real-world iCodon thermostability dataset, shows that HELM also remains more robust and superior to XE, for extreme sequence lengths and high GC content. Scaling experiments are also presented in Appendix Sec. A.8 demonstrating that hierarchical encoding remains critical even at larger model sizes.

Table 2: Performance comparison of HELM vs. XE models. HELM models encoding mRNA hierarchy outperform non-hierarchical XE models across all downstream tasks and datasets, highlighting the importance of hierarchy as a strong biological prior. Bold indicates the best performing model.

| Model | Ab1 | Ab2 | MLOS | iCodon | Tc-Riboswitches | mRFP | COV-19 Vaccine |
|---|---|---|---|---|---|---|---|
| Transformer XE (MLM) | 0.748 | 0.599 | 0.653 | 0.503 | 0.569 | 0.753 | 0.801 |
| Transformer HELM (MLM) | **0.767** | **0.609** | **0.701** | **0.525** | **0.626** | **0.822** | **0.833** |
| Transformer XE (CLM) | 0.752 | 0.597 | **0.611** | 0.498 | 0.531 | 0.815 | 0.787 |
| Transformer HELM (CLM) | **0.760** | **0.614** | 0.592 | **0.529** | **0.619** | **0.849** | **0.789** |

**When and why learning hierarchical mRNA encoding is most effective?** Hierarchical HELM modeling improves the performance in all predictive tasks we considered. However, the extent of improvement varies across datasets. Greater gains are seen in MLOS, Tc-Riboswitches, and mRFP datasets, while Ab1, Ab2, and iCodon show smaller improvements (see Table 2). We hypothesize that this variation is driven by synonymous codon usage bias which indicates the preference for certain synonymous codons (coding for the same amino acid) to be used more frequently than others due to factors like gene expression, tRNA availability, or evolutionary pressures (Sharp & Li, 1987; Parvathy et al., 2022). We hypothesize that datasets with more pronounced synonymous codon usage bias benefit more from learning with hierarchy as HELM treats amino acids as parent nodes and their synonymous codons as child nodes, enabling the model to learn codon usage bias by capturing preferences at both the amino acid and codon levels.

To evaluate synonymous codon usage bias for any dataset, we first calculate the distribution of synonymous codons for each amino acid and then compute the entropy of codon usage for the five most frequent amino acids. Lower entropy indicates less uniform codon distribution hence the preference towards specific codons or codon bias. Fig. 2 and Appendix Table 5 demonstrate that MLOS, Tc-Riboswitches, and mRFP datasets have lower entropy, suggesting stronger codon bias, which aligns with the larger improvements seen in these datasets using HELM models.

In addition to entropy-based metric, we extend our analysis of codon usage bias by looking at Codon Pair Bias (CPB), which assesses the over- or under-representation of codon pairs or higher-order k-mers within a sequence compared to a background distribution (Coleman et al., 2008). Each codon k-mer is assigned a weight based on the log-ratio of its observed to expected frequency. The overall CPB score for a sequence is the mean of these weights, with positive scores indicating over-represented k-mers and negative scores reflecting under-represented ones. Fig. 3 shows average CPB scores for each dataset, reinforcing findings from the entropic analysis and reinforcing the hypothesis that HELM's hierarchical approach is particularly beneficial for datasets with pronounced codon usage biases, highlighting its potential for improved mRNA modeling.

## 4.3 HELM IMPROVES GENERATIVE MRNA SEQUENCE DESIGN

Our HELM-CLM models are inherently generative and can be used for mRNA sequence generation. We evaluate their ability to produce biologically relevant and diverse mRNA sequences through two experiments: Frechet Biological Distance and functional properties preservation.

**Evaluating sample quality of generated mRNA** We adopt the Frechet Biological Distance (FBD) metric (Stark et al., 2024) to assess the quality of generated sequences. FBD is a stan-

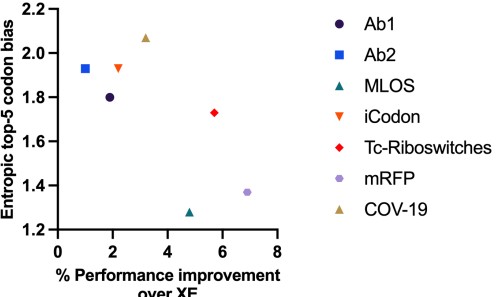 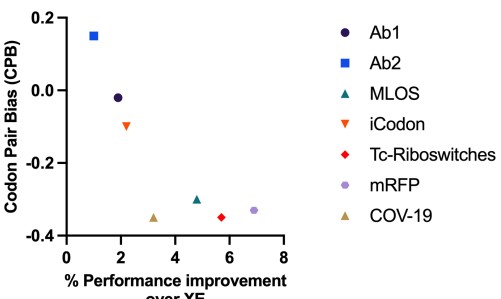

Figure 2: Average entropy of synonymous codon distributions correlates with HELM model performance improvement. Lower entropy, indicating stronger codon bias, is observed in datasets like MLOS, Tc-Riboswitches, and mRFP, where HELM shows greater improvements over XE models.

Figure 3: Relationship between Codon Pair Bias (CPB) and % improvement over XE. Datasets with more negative CPB values (indicating stronger codon usage bias) tend to exhibit greater improvements with HELM.

dard generative evaluation metric which quantifies the similarity between generated and real sample distributions, and is used for biological sequences. Our experiment uses XE and HELM CLM models to generate 2000 mRNA sequences encoding antibody D and J regions, conditioned on signal peptide and V regions from the OAS hold-out dataset. Generated and real mRNAs are translated to protein sequences, from which we extract embeddings using the ESM-2 model (Lin et al., 2023). FBD is computed as the Wasserstein distance between Gaussian distributions fitted to these embeddings, with lower scores indicating better alignment with real data. We focus on amino acid (and hence protein) level FBD since the encoding hierarchy reflects the codon-amino acid relationship.

In the absence of any other existing generative models for mRNA, we benchmark the HELM model against the non-hierarchical baseline trained with vanilla cross-entropy (XE) loss. As an additional control, we establish a random baseline by generating sequences randomly, following the approach of Stark et al. (2024). The random baseline serves as a control to assess the significance of the sequence generation capabilities of both our HELM model and the XE baseline.

Results in Fig. 4 demonstrate that both the HELM model and the XE baseline significantly outperform the random baseline, indicating that both models generate mRNA sequences that are meaningfully aligned with real data distributions. Unsurprisingly, the generative performance varies with generation temperature, with higher temperatures leading to greater diversity but worse FBD scores. Notably, HELM consistently achieves better FBD scores than XE across all temperatures, suggesting it produces sequences more representative of real mRNA data while maintaining generation diversity. These results demonstrate HELM's effectiveness in both predictive and generative tasks, highlighting its potential for improved mRNA sequence design.

**Evaluating functional properties of generated mRNA**  We assess the models' ability to generate sequences that maintain functional properties across six datasets from our downstream predictive tasks (Sec. 4.1.2). For each dataset, we select up to 2000 sequences, condition the models on the first third of each sequence, and task them with generating the remaining two-thirds. To evaluate the retention of functional properties in generated sequences, we employ pre-trained property prediction models to estimate task-specific labels, following the protocol established in Stark et al. (2024); Avdeyev et al. (2023). We use Mean Squared Error (MSE) between predicted labels of generated sequences and true labels of corresponding real sequences as our performance metric, with lower MSE indicating better preservation of functional properties.

Fig.5 illustrates the relative improvement in MSE for HELM models compared to non-hierarchical XE models across all datasets. HELM consistently outperforms XE, with improvements ranging from 2% to 31%. This demonstrates that HELM's hierarchical encoding better preserves functional properties across diverse tasks and datasets. Notably, we observe greater improvements for MLOS, Tc-Riboswitches, and mRFP datasets, which aligns with the stronger synonymous codon usage bias observed in these datasets (as discussed in Sec.4.2). These results further underscore HELM's effectiveness in capturing and preserving the biological nuances of mRNA sequences during generation.

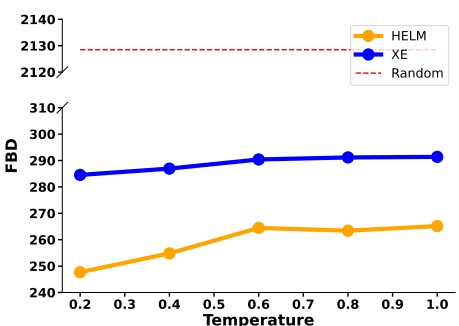
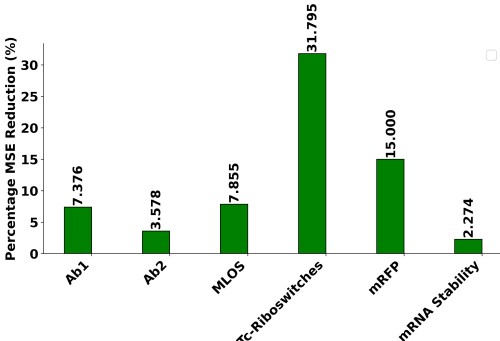

Figure 4: FBD comparison of generative HELM, XE, and random models across varying temperature. Higher temperatures increase diversity but worsen FBD scores. HELM consistently beats XE, suggesting better alignment with real mRNA data while maintaining diversity.

Figure 5: Percentage MSE reduction: HELM vs XE models. MSE compares the predicted properties of generated sequences to the predicted properties of original sequences. HELM consistently outperforms non-hierarchical models, indicating better retention of key mRNA properties in generated sequences.

### 4.4 HELM IMPROVES AB-MRNA SEQUENCE REGION ANNOTATION

We evaluate HELM's performance in annotating specific regions within antibody-encoding mRNA sequences. This task is crucial for understanding immune diversity, guiding therapeutic antibody development, and studying immune-related diseases (Briney & Burton, 2018). Currently, no machine learning-based tool exists for this specific annotation task.

We use a hold-out set of 2000 antibody-encoding sequences from curated OAS data, previously employed in our clustering and generative experiments. The objective is to predict whether each nucleotide belongs to one of four regions: signal peptides, V, DJ, or constant regions, effectively annotating the entire antibody-encoding sequence. Region labels are curated from the OAS database.

Appendix Table 9 shows that HELM significantly improves annotation accuracy compared to the non-hierarchical XE model, particularly when using the Masked Language Modeling (MLM) objective. This improvement can be attributed to two factors. First, HELM's hierarchical prior better captures the inherent structure of mRNA sequences, where certain regions are functionally and structurally dependent on others. Second, the bidirectional attention in MLM more effectively captures the context needed for precise sequence annotation, which is crucial for identifying distinct regions where both upstream and downstream nucleotide dependencies are important. For details see (Sec. A.9).

## 5 CONCLUSIONS

In this work, we highlight the importance of biological hierarchy in mRNA language modeling and show that standard natural language modeling approaches fall short in capturing the biological structure of mRNA sequences. We introduce HELM, a method to incorporate mRNA's hierarchical structure into bio-language model pre-training. By modulating loss based on codon hierarchy, HELM aligns the learning process with mRNA's inherent biological structure. In addition, we provide a consistent comparison of various tokenization methods, pre-training strategies, and model architectures to guide future mRNA LM development. HELM consistently outperforms non-hierarchical models by an average of 8% across six diverse downstream property prediction tasks with the biggest improvement observed on datasets with pronounced synonymous codon bias. Moreover, HELM models demonstrate significantly better generative capabilities than the non-hierarchical generative LMs enabling generation of more diverse and plausible mRNA sequences.

One limitation of our work is learning hierarchical relationships in Euclidean space, which may limit the model's ability to fully leverage the hierarchical prior. In contrast, hyperbolic spaces are naturally better suited for capturing hierarchical relationships, as they can effectively model the exponential growth of tree-like structures. This can be explored in future work to improve hierarchical learning and further enhance mRNA sequence modeling.

## REPRODUCIBILITY STATEMENT

We are committed to the transparency and reproducibility of our findings. Code, datasets, and model weights will be made publicly available to the research community, allowing others to reproduce, verify, and extend our work. Additionally, to ensure reproducibility, we have also provide detailed descriptions of model architecture, hyperparameters, training and inference procedures in Appendix Sections A.2 and A.3 with additional details on experimental information.

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

## A  APPENDIX

### A.1  CODON HIERARCHY

Figure 6 shows the formalized codon hierarchy used in Helm pre-training. The root node $R$ of this tree represents a node corresponding to all codons. This root node has two immediate child nodes: coding codons $C_{\text{coding}}$ and non-coding codons $C_{\text{non-coding}}$. The non-coding codons $C_{\text{non-coding}}$ further branch into two child nodes: the start codon node $C_{\text{start}}$ and the stop codon node $C_{\text{stop}}$. The start codon node $C_{\text{start}}$ has a single leaf node corresponding to the codon AUG. The stop codon node $C_{\text{stop}}$ has three leaf nodes corresponding to the codons UAA, UAG, and UGA. On the other side of the hierarchy, the coding codons $C_{\text{coding}}$ are organized into multiple child nodes, each corresponding to a specific amino acid $A_j$. Each amino acid node $A_j$ has a number of child nodes, each corresponding to a synonymous codon that encodes for $A_j$. Let $\text{Codons}(A_j) = \{C_{i_1}, C_{i_2}, \ldots, C_{i_m}\}$ denote the set of all synonymous codons for the amino acid $A_j$. These synonymous codons preserve the same amino acid, allowing for degeneracy in the genetic code. The complete hierarchical structure is thus represented by the tree $H = (V, E)$, where $V$ includes the root node, all internal nodes (corresponding to coding and non-coding codons, start/stop codons, and amino acids), and all leaf nodes (corresponding to individual codons). The edges $E$ define the hierarchical relationships between these nodes.

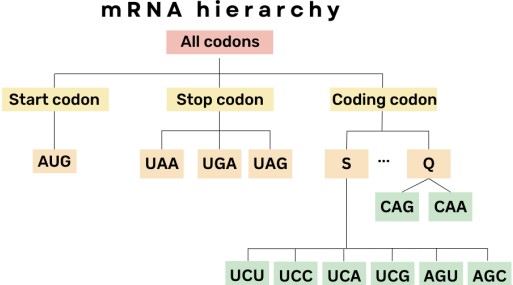

Figure 6: Formalized tree of the codon hierarchy used for pre-training HELM models

### A.2  PRE-TRAINING DETAILS

We experimented with GPT-2, Mamba, and Hyena architectures pre-trained on a curated dataset. GPT-2 was trained with both MLM and CLM objective, while Mamba and Hyena were trained exclusively with CLM, given that Mamba is specifically designed for Causal Language Modeling, and recent findings show that CLM is superior to MLM for representation learning (Nguyen et al., 2024b). The models are pre-trained with either vanilla cross-entropy loss (vanilla XE) or hierarchical cross-entropy loss (HXE), also referred to as the HELM model (See Sec. 4.1 in main text).

With codon tokenization, the maximum context length for pre-training is 444 tokens, equivalent to the longest sequence of 1332 nucleotides in the dataset. However, the models are designed to support sequences up to 2048 tokens, thanks to a positional embedding layer that scales to 2048 positional embeddings, offering flexibility for future use with longer sequences without architectural changes. All models were pre-trained for 40 epochs on 8 Nvidia A100 GPUs. For downstream tasks, a 1 million-parameter TextCNN(Kim, 2014) head was used for fine-tuning, following common probing techniques(Marquet et al., 2022; Chen et al., 2024; Outeiral & Deane, 2024; Harmalkar et al., 2023).

All models are designed to have approximately 50 million parameters. GPT-2 used 10 layers with a hidden size of 640 and an intermediate size of 2560. For position embedding, we follow the strategy used in GPT-2. We employ an embedding layer and add the embedded positional embeddings to the token embeddings. This approach is consistent across all models in our study.

Although GPT-2 supports CLM by design, we use attention without causal masking for creating MLM variant of it. We employ the AdamW optimizer with a learning rate scheduler using linear warmup and cosine decay. Initially, we use a learning rate of 1e-3 for all models. For hierarchical HELM models trained with HXE loss, we reduced the learning rate to 1e-4 for all models to accom-

modate the smaller scale of the loss compared to XE. We experimented with three different $\alpha$ values for HXE: 0.2, 0.4, and 0.6 (See Sec. 3 in main text for details) with $\alpha = 0.2$ performing the best overall.

All models are trained using 8 NVIDIA A100 GPUs, each with 80GB of GPU memory. We utilized mixed-precision training (AMP) to improve computational efficiency. The distributed training setup allowed us to effectively use a batch size of 1024 (128 per GPU across 8 GPUs).

We initially experimented with longer training durations but found no significant improvement beyond 40 epochs. Therefore, all models are trained for 40 epochs to ensure consistent comparison and efficient use of computational resources. The hyperparameters used for training are summarized in the following table:

Table 3: Hyperparameters for trained LM Models

| Hyperparameter | GPT-2 | Mamba | Hyena |
|---|---|---|---|
| Number of layers | 10 | 40 | 7 |
| Hidden size | 640 | 256 | 768 |
| Intermediate size | 2560 | 1024 | 3072 |
| Batch size | 1024 | 1024 | 1024 |
| Learning rate (XE) | 1e-3 | 1e-3 | 1e-4 |
| Learning rate (HXE) | 1e-4 | 1e-4 | 1e-4 |
| Minimum learning rate | 1e-5 | 1e-5 | 1e-6 |
| Weight decay | 0.1 | 0.1 | 0.1 |
| Number of epochs | 40 | 40 | 40 |
| Vocabulary size | 70 | 70 | 70 |

## A.3 DOWNSTREAM TASK FINETUNING DETAILS

To evaluate the effectiveness of our pre-trained LMs on downstream tasks, we employ probing approach (Marquet et al., 2022; Chen et al., 2024; Outeiral & Deane, 2024; Harmalkar et al., 2023) using TextCNN (Kim, 2014). This allows us to assess the quality and transferability of the learned representations across various downstream predictive tasks.

In our implementation of TextCNN, we set the embedding size to 640 to match the hidden size of our models, ensuring consistency across all probing experiments. The convolutional layer size is fixed at 100, which we find to provide a good balance between model capacity and computational efficiency. To focus solely on evaluating the quality of the learned representations, we freeze the weights of the pre-trained LMs, using them as fixed feature extractors to generate embeddings for the probing experiments. We conduct a comprehensive hyperparameter search to optimize the performance of the TextCNN probe on each downstream task. Our search strategy employs a grid search methodology, exploring a range of learning rates and batch sizes. Specifically, we investigate learning rates of 3e-4, 1e-4, and 1e-5, which span a reasonable range for finetuning neural networks. For batch sizes, we examine values of 8, 16, 32, and 64, allowing us to balance between update frequency and the stability of gradient estimates. The combination of three learning rates and four batch sizes results in twelve distinct configurations for each downstream task. We train the TextCNN probe using each of these configurations and evaluate their performance on a hold-out validation set to choose the best hyperparameters. For our final results, we report the performance as measured on a separate test set. This approach ensures that we are comparing the most optimized versions of each probing model, providing a fair assessment of the underlying representations learned by our pre-trained models. For all datasets, whether they come with predefined splits or not, we maintained a consistent data partitioning strategy. We divide the data into training, validation, and test sets with ratios of 0.75, 0.15, and 0.15, respectively. This split ensures a substantial portion of data for training while allocating sufficient samples for validation and testing. In cases where the datasets for downstream tasks does not come with predefined train-validation-test splits, we implement a robust splitting strategy. We randomly divide the data into training, validation, and test sets, maintaining consistent proportions across all tasks. To account for potential variability introduced by this random splitting, we repeat the entire process using three different random seeds. The reported results for these tasks represent the average performance across these three splits, providing a more reliable estimate of the model's

generalization capabilities. This comprehensive finetuning and evaluation process allows us to systematically and fairly assess the transferability and quality of the representations learned by our pre-trained models.

## A.4 TOKENIZATION STRATEGY EXPERIMENTS

We conducted experiments to compare the performance of different tokenization strategies on various mRNA datasets, including nucleotide-level, 6-mer, and codon-level tokenization. The experiments were performed using a 50M parameter GPT-2 model with a Masked Language Modeling (MLM) objective. Other experimental setups, such as data processing, training pipelines, and evaluation metrics, were kept consistent across all models to ensure a fair comparison of tokenization strategies.

### A.4.1 TOKENIZATION STRATEGIES

- **Nucleotide-level Tokenization:** This strategy treats each nucleotide (A, C, G, U) as a single token, resulting in a vocabulary size of 4.

- **6-mer Tokenization:** In this approach, sequences are split into overlapping or non-overlapping chunks of six nucleotides, generating a more complex vocabulary.

- **Codon-level Tokenization:** Codon-level tokenization represents sequences as triplets of nucleotides, resulting in a vocabulary of 64 codons, corresponding directly to amino acids.

### A.4.2 RESULTS

Table 4 presents a comparison of the performance of different tokenization strategies across several mRNA datasets in terms of Spearman rank correlation (higher the better). Codon-level tokenization generally performs best across most datasets, except for mRFP, where 6-mer tokenization outperformed codon-level tokenization.

Table 4: Comparison of tokenization strategies across different mRNA datasets. Codon-level tokenization generally performs best, with 6-mer outperforming on mRFP.

| Dataset | Nucleotide-level | Codon-level | 6-mer |
|---------|------------------|-------------|-------|
| Ab1 | 0.699 | **0.747** | 0.733 |
| Ab2 | 0.569 | **0.599** | 0.582 |
| MLOS | 0.528 | **0.654** | 0.625 |
| Tc-Riboswitches | 0.533 | **0.570** | 0.562 |
| mRFP | 0.820 | 0.754 | **0.857** |

The results show that codon-level tokenization captures the biological significance of mRNA sequences and is more suitable for most tasks. However, on MRFP and iCodon datasets, 6-mer tokenization yielded better performance.. The 50M parameter GPT-2 model with the MLM objective was used in all experiments, and all other experimental setups remained the same to facilitate fair comparison. For further experimental details and setup, please refer to this appendix.

## A.5 CODON USAGE BIAS ANALYSIS

### A.5.1 ENTROPIC CODON BIAS

Table 5 reveals that MLOS, Tc-Riboswitches, and mRFP datasets exhibit lower entropy values, indicating stronger codon usage bias compared to Ab1, Ab2, and iCodon datasets which have higher entropy and more uniform codon distributions. This observation aligns with our hypothesis, as HELM models achieve greater improvements in datasets with skewed codon usage, suggesting that hierarchical learning better captures and leverages these codon biases for enhanced accuracy.

Table 5: Average entropy values of synonymous codon distributions for the top five most frequent amino acids in each dataset reveal varying degrees of codon usage bias. Datasets like MLOS, Tc-Riboswitches, and mRFP, with lower entropy values, exhibit stronger codon usage bias, correlating with the enhanced performance of hierarchical HELM models. In contrast, datasets like iCodon and COV-19 Vaccine show relatively weaker codon usage bias, with Ab1 and Ab2 exhibiting the least bias, aligning with the comparatively smaller yet significant performance gains of hierarchical HELM models on these datasets (see Fig. 2 and Table 2).

| Dataset | Average Entropy (top-5) |
| --- | --- |
| Ab1 | 1.80 |
| Ab2 | 1.93 |
| MLOS | 1.28 |
| iCodon | 1.93 |
| COV-19 Vaccine | 2.07 |
| Tc-Riboswitches | 1.73 |
| mRFP | 1.37 |

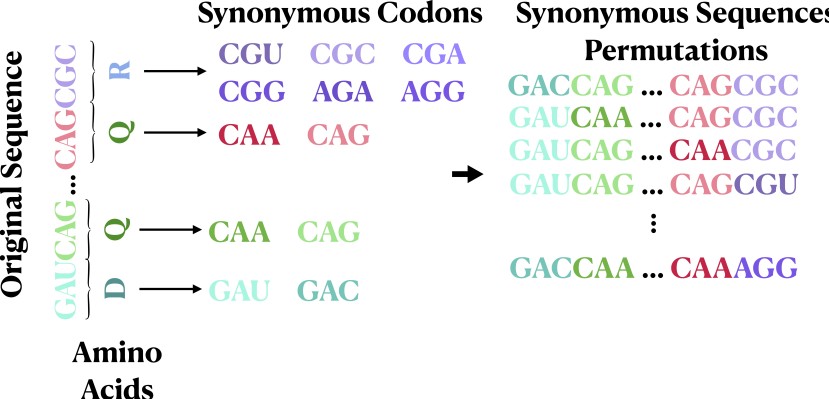

Figure 7: Illustration of the experiment designed to validate the hierarchy encoding quality in HELM models. The experiment involves generating synonymous mRNA sequences that code for the same protein by replacing codons with their synonymous alternatives. Hierarchical models should cluster these sequences tighter than non-hierarchical models.

## A.6    EVALUATING HIERARCHICAL ENCODING THROUGH SYNONYMOUS SEQUENCE CLUSTERING

To assess how effectively HELM captures the hierarchical relationships between amino acids and their synonymous codons in learned representations, we conduct a clustering analysis. We hypothesize that hierarchy-aware models should group biologically synonymous mRNAs (those coding for the same protein) more tightly than non-hierarchical models. To this end, we conduct experiment on a holdout set of 20000 curated OAS mRNA sequences not used in pre-training. For each sequence, we create 100 synonymous sequences by replacing codons with synonymous alternatives that code for the same amino acid, resulting in nucleotide-level variations but coding for the same protein sequences (see Fig. 7). We extract embeddings from both HELM and XE models and apply k-means clustering. The results, summarized in Table 6, support our hypothesis as HELM models produce significantly tighter clusters as measured by the Silhouette score (Rousseeuw, 1987), confirming their superior ability to learn and represent the hierarchical structure of mRNA sequences. This clustering behavior confirms HELM's ability to effectively learn these hierarchical relationships.

## A.7    IMPACT OF THE CHOICE OF $\alpha$ FOR HXE LOSS

In this section, we present a detailed analysis of the impact of the hyperparameter $\alpha$ on the performance of models trained using the Hierarchical Cross-Entropy (HXE) loss. The $\alpha$ parameter in

Table 6: Clustering performance metrics comparing hierarchical HELM models with vanilla XE models. Bold indicates better clustering.

|  | MLM Silh. ($\uparrow$) | CLM Silh. ($\uparrow$) |
|---|---|---|
| XE | 0.74 | 0.68 |
| HELM | **0.91** | **0.83** |

the HXE loss plays a crucial role in weighting the importance of hierarchical relationships during model training. Specifically, $\alpha$ controls how much weight is given to mistakes that violate the hierarchical structure of mRNA sequences, where codons are organized based on their synonymous relationships.

As explained in Sec. 3, the HXE loss leverages the hierarchical nature of mRNA sequences by applying differential penalties to the prediction errors based on their position within the hierarchy. The weighting function $\lambda(C)$ is $\lambda(C) = \exp(-\alpha h(C))$, where $h(C)$ is the height of the node $C$ and $\alpha > 0$, determines the extent to which errors are penalized. Higher values of $\alpha$ increase the penalty for mistakes higher up in the hierarchy (e.g., confusing codons that code for different amino acids), while lower values of $\alpha$ result in a more uniform penalty across the hierarchy.

Table 7: Ablation study on HXE loss $\alpha$ across downstream tasks

| Hyperparameter | Ab1 | Ab2 | iCodon | mRFP | Tc-Riboswitches |
|---|---|---|---|---|---|
| *Transformer HELM (MLM)* | | | | | |
| $\alpha = 0.2$ | 0.767 | 0.609 | 0.525 | 0.822 | 0.626 |
| $\alpha = 0.4$ | 0.767 | 0.608 | 0.511 | 0.831 | 0.653 |
| $\alpha = 0.6$ | 0.762 | 0.599 | 0.508 | 0.796 | 0.582 |
| *Transformer HELM (CLM)* | | | | | |
| $\alpha = 0.2$ | 0.760 | 0.614 | 0.529 | 0.849 | 0.619 |
| $\alpha = 0.4$ | 0.762 | 0.608 | 0.527 | 0.841 | 0.551 |
| $\alpha = 0.6$ | 0.752 | 0.592 | 0.526 | 0.798 | 0.580 |

To assess the impact of $\alpha$ we evaluated the performance of the transformer models trained with the HXE loss using three different values of $\alpha$: 0.2, 0.4, and 0.6. These values were chosen to explore a range of hierarchical weighting, from mild to strong penalties for hierarchical errors. Table 7 summarizes the performance of the models on each downstream task. We observe the following trends:

- $\alpha$ =0.2: At this lower value of $\alpha$, the model applies relatively mild penalties for hierarchical mistakes. The performance across most tasks is stable, indicating that even a modest incorporation of hierarchical weighting can enhance the model's performance by aligning with the biological structure of mRNA sequences.

- $\alpha$ =0.4: Increasing $\alpha$ to 0.4 generally results in improved performance for MLM, particularly for the mRFP and Tc-Riboswitches tasks. This suggests that at this level, the model benefits from a stronger alignment with the hierarchical structure, which is critical for these tasks. However, the performance on some tasks (e.g., Ab1 and Ab2) remains stable, indicating that the benefit of increased $\alpha$ might be task-dependent.

- $\alpha$ =0.6: When $\alpha$ is further increased to 0.6, we observe a decline in performance for several tasks, including mRFP and Tc-Riboswitches. This suggests that an overly strong hierarchical penalty can restrict the model's flexibility, leading to overfitting to the hierarchical relationships and a loss of generalization across diverse tasks.

Thus, our experiments show that a moderate value of $\alpha$ (0.2 to 0.4) generally yields the best results across the diverse set of tasks. This value strikes a balance between enforcing hierarchical structure and maintaining model flexibility.

## A.8 IS LEARNING HIERARCHY BENEFICIAL FOR LARGER SCALE MODELS?

In this section, we provide the details of our experiments with two scales of transformer-based models: 50 million (50M), and 100 million (100M) parameters. All models were trained using the non-hierarchical XE and our hierarchical HELM framework with a Causal Language Modeling (CLM) objective. The performance was evaluated on downstream mRNA property prediction tasks on five datasets introduced in main text.

### A.8.1 MODEL CONFIGURATIONS

- **50M Model:** This model consists of 10 transformer layers, each with a hidden size of 640 and intermediate size of 2560.

- **100M Model:** This model consists of 14 transformer layers, each with a hidden size of 768 and intermediate size of 3072.

All models were trained on the same mRNA dataset using the either XE or HELM-modified CLM objective, and we maintain a consistent training pipeline across all experiments.

### A.8.2 ENCODING HIERARCHY HELPS EVEN WITH SCALED UP MODELS

As shown in Appendix Table 8, the 100M HELM model still significantly outperforms its XE counterpart, maintaining the relative performance gap observed at the 50M scale. This confirms that XE models do not naturally capture hierarchical information, even with increased size, unless explicitly guided by hierarchical cross-entropy (HXE) loss. While further scaling to even larger models would provide additional insights, our computational constraints (pretraining time of 2 weeks for 100M models) highlight the practical advantage of HELM's hierarchical priors in enabling superior performance without requiring expensive scaling.

Table 8: Performance comparison of HELM vs. XE models for 50M and 100M parameter sizes. Spearman rank correlation is reported with bold indicating the best performance in each case.

| Model Size | Ab1 | Ab2 | MLOS |
|---|---|---|---|
| 50M XE | 0.752 | 0.597 | **0.611** |
| 50M HELM | **0.760** | **0.614** | 0.592 |
| 100M XE | 0.750 | 0.596 | 0.608 |
| 100M HELM | **0.765** | **0.619** | **0.664** |

Note that even with larger scales, encoding hierarchy is useful. However, scaling the model incurs longer pre-training times with 50M and 100M models taking approximately 49.5 hours and 82 hours respectively.

## A.9 AB-MRNA SEQUENCE REGION ANNOTATION EXPERIMENT

In this experiment, we approached the problem of annotating specific regions within b-mRNA sequences as a multi-class classification task. Specifically, we aimed to predict the start and end positions of critical regions within the sequence: V (variable), SP (signal peptide), DJ (diversity and joining), and C (constant) regions. To achieve this, we designed a model that predicted seven key positions: the start and end of the V region, the start and end of the SP region, the start and end of the DJ region, and the start of the C region. Notably, we did not predict the end of the C region since this boundary is trivially determined in the context of the sequences.

To ensure consistency across the dataset, we fixed the maximum number of nucleotides in the test dataset to 902. This allowed us to normalize the sequence length, ensuring that predictions were made for uniform positional slots across all samples. Each of the seven positions was treated as a separate prediction task within the classification framework.

As a comparison metric, we utilized the average accuracy across all predicted regions. By measuring the accuracy for each start and end position independently and averaging these results, we obtained a

Table 9: Annotation accuracy metrics comparing hierarchical HELM models with vanilla XE models. Bold indicates best performance.

|  | MLM Acc. ($\uparrow$) | CLM Acc. ($\uparrow$) |
|---|---|---|
| XE | 78.68 | 67.51 |
| HELM | **83.39** | **73.39** |

holistic view of the model's performance across the entire Ab-mRNA sequence. This metric enabled us to evaluate how well the model generalized across various regions and how effectively it could predict the boundaries of biologically relevant areas within the sequence.

### A.10 PROPERTY PREDICTION RESULTS IN TERMS OF OTHER METRICS

While we have reported property prediction performance in the main text in terms of the commonly used Spearman rank correlation metric, here we quantify results also in terms of Pearson correlation and $R^2$, see Table 10. Similar to the trend observed with Spearman correlation, our proposed HELM model outperforms baselines for all datasets in terms of these metrics as well.

Table 10: Performance comparison of HELM vs. XE models using Pearson and $R^2$ metrics. HELM models encoding mRNA hierarchy outperform non-hierarchical XE models across all downstream tasks and datasets, highlighting the importance of hierarchy as a strong biological prior. Bold indicates the best performing model.

| Model | Ab1 | Ab2 | MLOS | Tc-Riboswitches | mRFP | COV-19 Vaccine |
|---|---|---|---|---|---|---|
| Transformer XE (MLM) | 0.766 / 0.587 | 0.599 / 0.361 | 0.667 / 0.444 | 0.532 / 0.283 | 0.779 / 0.606 | 0.788 / 0.620 |
| Transformer HELM (MLM) | **0.793 / 0.629** | **0.603 / 0.364** | **0.719 / 0.516** | **0.601 / 0.361** | **0.847 / 0.717** | **0.811 / 0.657** |
| Transformer XE (CLM) | 0.757 / 0.573 | 0.596/ 0.356 | **0.563 / 0.316** | 0.485 / 0.235 | 0.880 / 0.774 | 0.762 / 0.580 |
| Transformer HELM (CLM) | **0.787 / 0.619** | **0.608 / 0.370** | 0.542 / 0.293 | **0.556 / 0.309** | **0.886 / 0.784** | **0.769 / 0.591** |

### A.11 ANALYSIS OF HELM VS XE WITH DIFFERENT SEQUENCE LENGTHS AND GC CONTENT

**Sequence Length Analysis** In this experiment, we divide the iCodon thermostability dataset into three sequence length categories: 30-1000, 1000-2000, and 2000-3000 nucleotides. These categories help analyze how model performance varies as sequence length increases. For mRNA vaccines, the typical length of sequences falls within the range of 1000-1500 nucleotides (Gunter et al., 2023). Thus, our analysis of sequences in the 2000-3000 nucleotide range covers longer sequences than typically observed in mRNA vaccine datasets.

For both non-hierarchical XE and hierarchical HELM models, performance decreases as the sequence length increases as shown in Appendix Table 11. This drop is due to the fact that our pre-training dataset includes sequences with a maximum length of approximately 1400 nucleotides. Hence, sequences longer than this range are less effectively represented by both XE and HELM models.

Despite the performance degradation, HELM models exhibit much less performance drop compared to XE models, particularly in the longer sequence categories as exhibited by Appendix Table 11. This suggests that HELM's hierarchical encoding mechanism helps maintain performance even with longer sequences.

**GC Content Analysis** We also categorize the iCodon thermostability dataset based on GC content into three ranges: GC $\leq$ 47, 47 < GC $\leq$ 55, and GC > 55. This classification closely aligns with widely recognized thresholds in the literature, where GC content below 45% is considered low, and above 56% is high.

Performance for both XE and HELM decreases with extremely high GC contents as shown in Appendix Table 12. Again, this is likely due to the fact that pre-training data mostly contains sequences with normal GC content range (GC < 50) and higher GC content sequences are less common. Once again, HELM models show superior performance compared to XE models across all GC content

Table 11: Thermostability prediction performance of XE and HELM models across different sequence length ranges shown with Spearman rank correlation (higher value indicates better performance). The performance of both models decreases with increasing sequence length, particularly in the 2000-3000 nucleotide range. HELM demonstrates more robustness to sequence length variation.

| Sequence Length Range | XE (CLM) | HELM (CLM) | Number of Sequences |
|---|---|---|---|
| 30-1000 | 0.523 | **0.530** | 23929 |
| 1000-2000 | 0.496 | **0.528** | 28943 |
| 2000-3000 | 0.468 | **0.514** | 12484 |
| **Sequence Length Range** | **XE (MLM)** | **HELM (MLM)** | **Number of Sequences** |
| 30-1000 | 0.520 | **0.531** | 23929 |
| 1000-2000 | 0.506 | **0.529** | 28943 |
| 2000-3000 | 0.475 | **0.506** | 12484 |

ranges, particularly for the high GC content range, where XE shows more pronounced performance drop than HELM.

Table 12: Thermostability prediction performance of XE and HELM models across different GC content ranges shown with Spearman rank correlation (higher value indicates better performance). Performance decreases for both models with higher GC content, but HELM remains more robust.

| GC Content Range | XE (CLM) | HELM (CLM) | Number of Sequences |
|---|---|---|---|
| $GC \leq 47$ | **0.568** | 0.562 | 21783 |
| $47 < GC \leq 55$ | 0.440 | **0.485** | 21803 |
| $GC > 55$ | 0.517 | **0.529** | 21770 |
| **GC Content Range** | **XE (MLM)** | **HELM (MLM)** | **Number of Sequences** |
| $GC \leq 47$ | 0.557 | **0.570** | 21783 |
| $47 < GC \leq 55$ | 0.446 | **0.496** | 21803 |
| $GC > 55$ | 0.516 | **0.524** | 21770 |

## A.12 ADDITIONAL BASELINE

Table 13: Performance comparison of HELM models with hierarchical cross-entropy (HXE) loss vs. baseline models with separate codon-type embeddings using XE loss. Spearman rank correlation is reported. Bold indicates the best performance.

| Model | Ab1 | Ab2 | MLOS | Tc-Riboswitches | mRFP | COV-19 Vaccine |
|---|---|---|---|---|---|---|
| Baseline Sep. codon embedding (MLM) | 0.736 | 0.579 | 0.457 | 0.572 | 0.810 | 0.780 |
| HELM (MLM) | **0.767** | **0.609** | **0.701** | **0.626** | **0.822** | **0.833** |
| Baseline Sep. codon embedding (CLM) | 0.729 | 0.572 | 0.521 | 0.599 | 0.825 | 0.781 |
| HELM (CLM) | **0.760** | **0.614** | **0.592** | **0.619** | **0.849** | **0.789** |

We also implement a simpler baseline by introducing separate embeddings for different types of codons (e.g., [start], [stop], [S], [Q]) and combining these with the token embeddings as input, while using the standard cross-entropy (XE) loss without explicitly modeling the hierarchy. We train both MLM and CLM models with this setup and evaluated their performance on downstream property prediction datasets introduced in main text. The results, reported in Table 13, demonstrate that HELM models, which explicitly incorporate hierarchy, consistently outperform this baseline on all datasets irrespective of the pre-training strategy (MLM or CLM). This further highlights the importance of hierarchical modeling in achieving robust performance across diverse datasets.

### A.13 GENERALIZATION OF MODELS WITH AB PRE-TRAINING TO GENERAL MRNA SEQUENCES

As reported in the main text (Table 2), both XE and HELM models generalize well to various types of mRNA sequences even though they have been pre-trained on the Ab-mRNA dataset. This generalization can be attributed to two key mechanisms:

- Conserved RNA structural features: RNA molecules, whether synthetic or natural, share conserved structural motifs like loops, bulges, and stems that are critical for stability and function (Saito & Inoue, 2009).
- Shared molecular interactions: as antibody mRNAs often encode features related to interactions with nucleic acid-like antigens, creating overlap in sequence characteristics with other types of RNA molecules (Alberts et al., 2002).

These factors contribute to generalization of features learned from Antibody-mRNA to other types of mRNA observed in our experiments.

### A.14 PRE-TRAINING DATA CURATION AND STATISTICS

As discussed in Sec. 4 in main text, we ensure that the mRNA data used for pre-training our models in this study is of the highest quality, with careful attention to preserving the biological diversity and representativeness of the sequences. We conduct statistical analyses to confirm that the curated dataset maintains a balanced representation of gene types, comparable to the original OAS human dataset and did not introduce any biases in the gene representation.

First, we filtered sequences based on the *ANARCI status* annotation in the dataset, excluding those with unusual residues, indels, truncations, or a lack of conserved cysteines, which are often problematic. We further refined our selection by focusing only on sequences with a V and J identity greater than 0.7, ensuring a high degree of similarity to known reference sequences. Additionally, we retained only those sequences labeled as *productive* and *complete vdj*, indicating they are fully functional and contain complete variable, diversity, and joining regions. For the purpose of this study, we restricted the dataset to human sequences only. Recognizing that the paired sequences in the database are significantly fewer in number compared to unpaired ones, we unpaired these sequences to increase the dataset's size and diversity. We then performed sequence similarity analysis separately on paired and unpaired sequences to eliminate redundancy with a threshold of 0.5. To maintain a balanced representation, we manually adjusted the number of heavy and light chains. Finally, we compared the gene frequencies of heavy and light chains in our curated dataset against the original human OAS dataset to ensure that our curation process did not introduce artificial overrepresentation or underrepresentation of any gene types. This process yields 15.3 million sequences with 7.7 million heavy and 7.6 million light chains each.

The following figures present the distribution of gene types for heavy and light chains in both the original and curated datasets, as well as a breakdown of heavy and light chain numbers, including Kappa (K) and Lambda (L) isotypes.

