# OpenReview forum: "HELM: Hierarchical Encoding for mRNA Language Modeling"
_ICLR.cc/2025/Conference — ICLR 2025 Poster_

### Official Review · Reviewer_SJYV · 2024-11-03

**Soundness:** 3
**Presentation:** 3
**Contribution:** 2
**Rating:** 6
**Confidence:** 3

**Summary:**

The paper introduces a pre-training strategy, **Hierarchical Encoding for mRNA Language Modelling (HELM)**, that incorporates the hierarchical structure of mRNA codons into language model training. By modulating the model's loss based on codon synonymity, HELM aligns learning with the biological structure of mRNA. Results show HELM outperforms non-hierarchical models by around 8% on various mRNA tasks, including antibody annotation and property prediction, while also enhancing generative capabilities for mRNA sequences.

**Strengths:**

- HELM integrates codon hierarchy into model training, capturing the inherent structure of mRNA data.
- Demonstrates a good performance improvement over existing models on multiple downstream tasks.
- The model can generate more biologically plausible and diverse mRNA sequences than existing approaches.

**Weaknesses:**

- The authors curate a dataset from OAS specifically to train their models. This may explain much of the performance gains observed in Table-1, as the model is trained and tested on data with similar statistical characteristics. An ablation study isolating the effect of the curated dataset would clarify the extent to which performance gains are due to the data source versus tokenization and architecture design.
- As the authors noted it, representing hierarchical relationships in Euclidean space might limit HELM’s ability to capture the full complexity of these structures, potentially affecting performance.
- Although the results are encouraging, they are primarily based on antibody and property prediction tasks, which could restrict the model's applicability to other mRNA-related contexts.
- Additionally, the strategy for splitting the data into training, validation, and test sets is crucial and should be detailed in the main paper. In sequence-based datasets, it is common practice to split data based on clusters after clustering sequences, which minimizes data leakage. The reported performance gains may be overstated if there is any leakage between the splits.


I am open to revisiting my score; however, I will wait for: (i) the authors' response to my comments, and (ii) feedback from other reviewers.

**Questions:**

- Are you planning to open-source the curated dataset ?

---

> ### Author Response · Authors · 2024-11-20
> **Response to review (and invitation for further discussion)**
>
> We thank the reviewer for their feedback and acknowledging the performance improvement of our method over existing models on multiple predictive and generative tasks. We address the valuable questions and comments raised by the reviewer point-by-point:
>
> 1. ### An ablation study isolating the effect of the curated dataset:
>     We thank the reviewer for raising this important point. To ensure a fair comparison, we had already reported the XE baseline (Table 2), which is trained on the same dataset, architecture, and tokenization as our HELM approach. The only difference is that XE employs MLM and CLM pre-training without hierarchical encoding. Using the same pre-training dataset, HELM outperforms XE by approximately ~4% on property prediction tasks (Table 2), ~6% on generative tasks (Figure 5), and ~5% on antibody region annotation tasks (Table 4). These consistent improvements clearly demonstrate the advantages of hierarchical hierarchical HELM pre-training. We have clarified this point in the main text (lines 363–365 and highlighted in blue).
>
> 2. ### Experiments primarily based on antibody and property prediction tasks:
>     We respectfully disagree with the assessment. Firstly, we wish to clarify that in addition to antibody sequences, our work provides the results on five diverse non-antibody datasets (iCodon, Tc-Riboswitches, COVID-19 mRNA Vaccine, MLOS, mRFP) from multiple various species on both predictive and generative tasks in Tables 1, 2 and Fig.5. Secondly, we want to emphasize that antibody-focused datasets/tasks are themselves extremely relevant for therapeutics, with many recent ML papers at top conferences [1,2,3] specifically targeting the antibody domain due to its significant real-world impact.
>
> 3. ### Strategy for splitting the data:
>     We appreciate the reviewer’s emphasis on careful data splitting to prevent the leakage. As the reviewer notices, a common practice is to split the data based on clusters to minimize data leakage. This is exactly the approach implemented for AB1, AB2, and OAS region annotation datasets where we first partition the datasets into clusters (LinClust, similarity threshold of 0.9), and then randomly split those clusters into training/validation and test clusters to prevent data leakage. We included the details of this procedure in the main text (lines 263-266). For all other datasets, pre-defined train, val, and test splits already exist from prior publications and we directly reuse them to ensure consistent comparison with prior works.
>
> 4. ### Representing hierarchical relationships in Euclidean space might limit HELM’s ability to capture the full complexity of these structures:
>     In the discussion section (lines 535-539), we openly acknowledged the limitation of Euclidean space to model more complex hierarchical relationships in data, and we acknowledged the potential benefit of more sophisticated methods such as modeling hierarchy in hyperbolic spaces. The latter is a highly non-trivial task that requires hyperbolic layers, Riemannian optimizers, and prototype embedding techniques. As the first stage towards advanced hierarchy modeling, this work aims to demonstrate the potential of hierarchy, establish a baseline, and provide an experimental basis for further research in this direction.
>
> 5. ### Are you planning to open-source the curated dataset?
>      Our group is fully committed to making all code, model weights, and data (including the curated dataset) publicly available to ensure the complete reproducibility of all the results. In alignment with our organization/institution’s policy, we are permitted to release code and data, only after the paper’s acceptance. To address this, we have added a formal declaration of reproducibility at the end of the main text (page 11, highlighted in blue), where we explicitly commit to releasing all materials necessary to reproduce this work. Furthermore, we had also provided all model training and inference details in the Appendix Sections A.2 and A.3.
>
> [1] Frey et al. Protein Discovery with Discrete Walk-Jump Sampling. ICLR. 2024.
>
> [2] Villegas-Morcillo et al. Guiding diffusion models for antibody sequence and structure co-design with developability properties. NeurIPS. 2023.
>
> [3] Zhu et al. Antibody Design Using a Score-based Diffusion Model Guided by Evolutionary, Physical and Geometric Constraints. ICML. 2024.

---

> ### Author Response · Authors · 2024-11-25
> **Friendly request for feedback on responses to your questions...**
>
> We greatly appreciate the time and effort you’ve put into reviewing our submission. As the discussion period comes to a close tomorrow, we kindly request your feedback regarding our detailed responses to your valuable comments and the revisions we’ve made.
>
> Given our thorough responses and the comprehensive improvements made to the manuscript, we hope you will consider that our contribution merits a score well above marginal acceptance, reflecting its broader impact and technical depth. We believe we have effectively addressed all concerns, demonstrating the wide applicability and methodological novelty of our work.
>
> Your insights and updated evaluations would be incredibly helpful for the final assessment of our submission. Thank you once again for your thoughtful review and support during this process.

---

> > ### Comment · Reviewer_SJYV · 2024-11-30
> >
> > Thank you for your response and the revisions to the paper. My apologies for the delayed reply. While I still find the technical contribution of the paper to be limited, I recognise that the curated dataset could be considered as a valuable contribution. I plan to raise my score to 5 to allow room for further discussion. That said, I’d like to highlight a specific point raised by another reviewer regarding the generalisability of antibody mRNA pre-training to natural RNA sequences. I’ve read your explanation about conserved RNA structural features and shared molecular interactions, but these are just your assumptions. Would it be feasible for you to conduct an experiment to provide evidence supporting your claims / assumptions?

---

> ### Author Response · Authors · 2024-12-01
> **Quantitative Analysis of mRNA Structural Motif Conservation Between Antibody and Non-Antibody mRNA Sequences to Quantitatively Address Reviewer's Question**
>
> We appreciate the opportunity to address this point regarding the generalizability of antibody (Ab) mRNA pre-training to other mRNA sequences. As detailed in our response to reviewer 9ttH also, our hypothesis is not speculative but is rooted in established biological literature, which demonstrates conserved mRNA structural features and shared molecular interactions across antibody and non-antibody mRNA types. Specifically, previous studies [1, 2] (which we also cited in response to the reviewer 9ttH's questions) have shown that both Ab and non-Ab mRNAs share similar structural motifs critical for stability and function, including conserved loops and bulges that contribute to secondary structure formation.
> To further substantiate this and respond to the reviewer’s query for an analysis on this, we have now conducted a detailed analysis that directly compares the mRNA secondary structures of one of our Ab mRNA datasets (Ab1) and four non-Ab mRNA datasets (Cov-19 Vaccine, Tc-Riboswitches, MLOS, mRFP) presented in the paper. For all these datasets, our model showed high predictive performance, making them ideal candidates for this analysis.
>
> ***Analysis Details***
> 1. Secondary Structure Prediction: We predicted RNA secondary structures for each sequence in the datasets using two widely used tools: EternaFold [3] and ViennaRNA [4]. The analysis considers structural motifs (e.g., loops and bulges), grouping contiguous unpaired bases together.
> 2. Motif-Based Similarity Metric: For each pair of sequences from the Ab and non-Ab mRNA datasets, we calculated a similarity score by comparing the size and position of unpaired motifs. A cumulative similarity score was generated by averaging pairwise scores across all comparisons, offering a single metric to quantify structural overlap between the datasets. The similarity scores range from 0 (no structural similarity) to 1 (perfect structural similarity).
> The similarity scores obtained using both prediction tools are summarized below:
>
> ### **Table 1: Motif Similarity Scores for Ab1 and Tc-Riboswitches dataset**
>
> | Metric                         | EternaFold Score | ViennaRNA Score |
> |--------------------------------|------------------|-----------------|
> | Mean Pairwise Motif Similarity | 0.40          | 0.45          |
>
> ---
>
> ### **Table 2: Motif Similarity Scores for Ab1 and Cov-19 Vaccine dataset**
>
> | Metric                         | EternaFold Score | ViennaRNA Score |
> |--------------------------------|------------------|-----------------|
> | Mean Pairwise Motif Similarity | 0.38            | 0.40           |
>
> ---
>
> ### **Table 3: Motif Similarity Scores for Ab1 and MLOS dataset**
>
> | Metric                         | EternaFold Score | ViennaRNA Score |
> |--------------------------------|------------------|-----------------|
> | Mean Pairwise Motif Similarity |      0.44       | 0.44          |
>
>
> ---
>
> ### **Table 4: Motif Similarity Scores for Ab1 and mRFP dataset**
>
> | Metric                         | EternaFold Score | ViennaRNA Score |
> |--------------------------------|------------------|-----------------|
> | Mean Pairwise Motif Similarity |      0.40       | 0.47           |
>
>
> Scores between 0.38 and 0.47, as observed in our study, indicate meaningful overlap in structural motifs, aligning with biological expectations for conserved secondary structure features across Ab and non-Ab mRNA types considered in our study, thus supporting the hypothesis of conserved mRNA structural features. The scores are consistent across two independent secondary structure prediction tools, reinforcing the robustness of the analysis. We will be happy to include this in our camera-ready version as well. While we believe this approach robustly addresses the reviewer’s query, we are still open to more specific suggestions from the reviewer for additional ways to further validate this.
>
> **Request for Score Consideration**
>
> We would also like to highlight that this analysis complements the extensive experimental validations and other investigations we already conducted to address the other questions of the reviewers. In light of the additional effort and clarity provided, we respectfully request that the reviewer considers the many experiments and results we provided for rebuttal in their evaluation and potentially increase their score to reflect the rigor and thoroughness of our proposed method and paper enhanced by valuable reviewer suggestions.
>
> [1] Synthetic biology with RNA motifs, Saito & Inoue, Int J Biochem Cell Biol, 2009
>
> [2] The Generation of Antibody Diversity, Alberts et al., Molecular Biology of the Cell. 4th edition. 2002
>
> [3] RNA secondary structure packages evaluated and improved by high-throughput experiments, Wayment-Steele, Nature Methods, 2022
>
> [4] ViennaRNA package 2.0., Lorenz et al.,  Algorithms for Molecular Biology, 6(1):26, 2011

---

> > ### Comment · Reviewer_SJYV · 2024-12-02
> >
> > Thanks for your response, I really appreciate it --- well done! I raised my score to 6.

---

> > > ### Author Response · Authors · 2024-12-02
> > > **Thank you for raising your score further!**
> > >
> > > Thank you for your review and valuable suggestions which have helped improve our paper. Also thank you for your encouragement and raising your score again!

---

### Official Review · Reviewer_eRzE · 2024-11-03

**Soundness:** 3
**Presentation:** 3
**Contribution:** 2
**Rating:** 6
**Confidence:** 3

**Summary:**

The paper introduces HELM, a hierarchical encoding approach tailored for mRNA language modeling that leverages the codon-level structure of mRNA sequences. Unlike traditional models, HELM incorporates a hierarchical cross-entropy loss function to align with the biological structure of mRNA, particularly focusing on synonymous codon usage and its functional implications. The model is evaluated on multiple mRNA datasets, demonstrating improvements in downstream prediction tasks, generative diversity, and sequence annotation accuracy over non-hierarchical baselines. Overall, HELM represents a biologically-informed advancement in mRNA modeling, showing enhanced performance on tasks relevant to protein synthesis and gene expression.

**Strengths:**

1. Biological Prior Integration: The hierarchical encoding strategy effectively incorporates biological knowledge of mRNA structure, particularly the synonymous codon usage, which enhances the model’s performance on property prediction and generative tasks.
2. Diverse Evaluations: HELM's performance is thoroughly evaluated across multiple tasks, including property prediction, generative sequence design, and antibody sequence region annotation, showcasing the model’s versatility and relevance to bioinformatics applications.
3. Improved Representational Quality: By aligning model training with the codon hierarchy, HELM captures the underlying biological structure more effectively, achieving better clustering of synonymous sequences and improved predictive accuracy in key bioinformatics tasks.

**Weaknesses:**

1. Lack of Scaling Experiments: The paper primarily employs models with 50M parameters, which is relatively small compared to large language models (LLMs) in NLP. This limitation raises concerns about the necessity of HELM’s hierarchical encoding, as larger models might naturally learn these hierarchical relationships without explicit design. Experiments with larger models could help clarify if the hierarchical loss function offers unique advantages or if it becomes redundant with increased scale.
2. Limited technical contribution:  The primary contribution of this work lies in adapting hierarchical cross-entropy (HXE) to the mRNA structure, which, while valuable for bioinformatics, represents a limited advancement from a machine learning perspective. The adaptation of HXE to this domain may not provide significant innovation to the broader ML community.
3. Baseline suggestion: A simpler way to capture the hierarchical structure could be to introduce separate embeddings for different types of codons and combine these with the token embeddings as input. For instance, embeddings such as [start], [stop], [S], and [Q] (as shown in Figure 6) could be used to represent codon types without modifying the loss function. This approach would provide a straightforward baseline for evaluating the benefits of the proposed HXE solution.

**Questions:**

See W3.

---

> ### Author Response · Authors · 2024-11-20
> **Response to review (and invitation for further discussion)**
>
> We thank the reviewer for their positive feedback and highlighting the efficient incorporation of biological knowledge in our model, leading to enhanced performance on predictive and generative tasks.
>
> 1. ### Lack of Scaling Experiments:
>     We appreciate the suggestion to investigate scaling effects beyond 50M parameters. We have now conducted additional experiments with 100M parameter models (comparable to other state-of-the-art RNA/mRNA language models such as RNA-FM and CodonBERT at 100M). Our results show that 100M hierarchical HELM models continue to outperform 100M non-hierarchical XE baselines, with the performance gap remaining consistent with what we observed at the 50M scale. These findings, reported in Appendix Sec. A.7 on page 21 and highlighted in blue, suggest that the advantage of HELM pre-training holds on larger scales as well.
>
>     | Model                 | AB1 | AB2 | MLOS|
>     |-----------------------|-----|-----|-----|
>     |Transformer XE (50M)   |0.752|0.598|0.611|
>     |Transformer HELM (50M) |0.761|0.614|0.592|
>     |Transformer XE (100M)  |0.751|0.597|0.608|
>     |Transformer HELM (100M)|0.766|0.619|0.665|
>
>
>
> 2. ### Limited technical contribution:
>     We want to emphasize that our work, to the best of our knowledge, is the first one to explicitly bridge known biological hierarchy and language models. With this work, we aim to demonstrate the significance of hierarchy, establish a hierachical baseline, and provide an experimental basis to foster future research in the direction of hierarchical bio-language models. From this perspective, we consider the technical simplicity of the proposed method as a strength. It enables seamless integration of biological hierarchy into any mRNA language model with minimal modifications while still considerably improving both predictive and generative ascpects of a model.
>
>
> 3. ### Baseline suggestion by reviewer:
>     We appreciate the constructive suggestion for the alternative baseline! We implemented this by training models with separate embeddings for different codon types combined with token embeddings as input, using standard (non-hierarchical) MLM and CLM objectives. As shown in our new results in Appendix A.11 (page 23, Appendix Table 13 and highlighted in blue), HELM models with explicit hierarchical learning still outperform this baseline across all datasets, regardless of whether MLM or CLM pre-training is used.
>
>     | Model                      | AB1              | AB2               | MLOS               | Tc-Riboswitches   | mRFP              | COV-19 Vaccine    |
>     |----------------------------|------------------|-------------------|--------------------|-------------------|-------------------|-------------------|
>     | Suggested Baseline (MLM)   | 0.736            | 0.579             | 0.457              |  0.572            |  0.810            | 0.780             |
>     |  HELM (MLM)                | **0.767**        | **0.609**         |  **0.701**         | **0.626**         | **0.822**         |  **0.833**        |
>
>     | Model                      | AB1              | AB2               | MLOS               | Tc-Riboswitches   | mRFP              | COV-19 Vaccine    |
>     |----------------------------|------------------|-------------------|--------------------|-------------------|-------------------|-------------------|
>     | Suggested Baseline (CLM)   |  0.729           | 0.572             | 0.521              |  0.599            |  0.825            | 0.781             |
>     |  HELM (CLM)                | **0.760**        | **0.614**         |  **0.592**         | **0.619**         | **0.849**         |  **0.789**        |

---

> ### Author Response · Authors · 2024-11-25
> **Follow-Up on Feedback and Clarifications**
>
> We greatly appreciate the time and effort you’ve put into reviewing our submission. As the discussion period comes to a close tomorrow, we kindly request your feedback regarding our detailed responses to your valuable comments and the revisions we’ve made.
>
> Given our thorough responses and the comprehensive improvements made to the manuscript, we hope you will consider that our contribution merits a score well above marginal acceptance, reflecting its broader impact and technical depth. We believe we have effectively addressed all concerns, demonstrating the wide applicability and methodological novelty of our work.
>
> Your insights and updated evaluations would be incredibly helpful for the final assessment of our submission. Thank you once again for your thoughtful review and support during this process.

---

> > ### Comment · Reviewer_eRzE · 2024-11-25
> > **Response by Reviewer**
> >
> > Thank you for your response. I have carefully reviewed the authors' reply and sincerely appreciate their additional efforts in addressing the scaling and including baselines. However, my primary concern regarding the marginal contribution remains unresolved. Therefore, I will maintain my original score, which is already a positive assessment.

---

### Official Review · Reviewer_9ttH · 2024-11-04

**Soundness:** 2
**Presentation:** 3
**Contribution:** 2
**Rating:** 5
**Confidence:** 4

**Summary:**

The paper introduces Hierarchical Encoding for mRNA Language Modeling (HELM), a novel pretraining strategy that incorporates the hierarchical codon structure of mRNA into language model training. It addresses the limitations of existing models that overlook codon synonymity, which can lead to suboptimal performance in mRNA tasks. HELM modifies loss functions to prioritize errors between different amino acids over synonymous codons, enhancing the model's alignment with biological realities. Evaluations show that HELM outperforms standard language model pretraining and other state-of-the-art RNA models by approximately 8% across six diverse downstream tasks, including property prediction and antibody region annotation, while using fewer model parameters. Additionally, HELM demonstrates improved generative capabilities, producing mRNA sequences that better align with true data distributions. Overall, HELM effectively captures the hierarchical nature of mRNA, leading to enhanced performance in both predictive and generative tasks.

**Strengths:**

1. This study proposed a novel approach for embedding biological hierarchical structures into language models to enhance interpretability and accuracy in biological sequence analysis.
2. A comprehensive technical evaluation spanning multiple model architectures was presented, benchmarked across diverse datasets and use cases.
3. Clear and precise presentation of technical methods and experimental setup were provided to facilitate reproducibility and further research.

**Weaknesses:**

1. The performance improvement seems more attributable to data selection than methodology: - HELM uses antibody mRNA while baselines use different data types (ncRNA, pre-mRNA, diverse organisms mRNA).
2.  The paper fails to justify why antibody mRNA pre-training would generalize to: - Viral RNA sequences - Riboswitch sequences.
3.  The evaluation methodology is insufficient: a. Over-reliance on single metric (Spearman correlation) b. No analysis across sequence lengths c. No biological interpretation of prediction errors d. The evaluation lacks comprehensive stress testing on edge cases (e.g., extreme sequence lengths, unusual GC contents, rare codon clusters), raising concerns about the model's reliability in challenging real-world scenarios.
4. Methodological issues: (1) α parameter (0.2-0.6) lacks biological basis (2) Oversimplified codon bias analysis (3) Insufficient validation of hierarchical structure's biological relevance.
5. Limited practical applications: (1) Too narrow focus on antibody sequences and expression (2) Critical applications remain untested: a. RNA vaccine design applications b. Abnormal or mutated sequences c. Real-world biological scenarios.

**Questions:**

1. What is the biological justification for the α parameter range selection?
2. How does the model handle sequences with unusual codon usage patterns? Such evaluation is crucial because unusual codon usage patterns could significantly impact translation efficiency and protein expression levels, yet these scenarios are not addressed in the current validation.

**Details Of Ethics Concerns:**

No.

---

> ### Author Response · Authors · 2024-11-20
> **Response to review and invitation for further discussion (Response 1 of 3)**
>
> We thank the reviewer for appreciating the novelty and highlighting the comprehensive evaluation of our method accross various architectures, tasks and datasets. We address all questions raised by the reviewer point-by-point grouping similar questions/comments together:
>
> 1. ### Performance improvement seems more attributable to data selection than methodology:
>     We thank the reviewer for raising this important point. To ensure a fair comparison, we had already reported the XE baseline (Table 2, page 8), which is trained on the same dataset, architecture, and tokenization as our HELM method. The only difference is that XE employs MLM and CLM pre-training without hierarchical encoding. Using the same pre-training dataset, HELM outperforms XE by approximately ~4% on property prediction tasks (Table 2), ~6% on generative tasks (Figure 5, page 10), and ~5% on antibody region annotation tasks (Table 4, page 9). These consistent improvements clearly demonstrate the advantages of hierarchical HELM pre-training. We have clarified this point in the main text (lines 363–365 highlighted in blue).
>
> 2. ### Why antibody mRNA pre-training generalizes to viral RNA sequences, riboswitch sequences:
>     The model can generalize to viral mRNA and riboswitch sequences because mRNA sequences, including antibody mRNA, share conserved structural and functional motifs shaped by their evolutionary process. Many antibody mRNA properties depend on their secondary structure elements such as loops and stems, which are similarly important for the stability and function of viral mRNA and riboswitches [1]. Additionally, antibodies frequently target nucleic acid-like antigens, which creates an overlap in the sequence features of antibody mRNA and other RNA-based molecules [1,2]. These factors support generalization of antibody mRNA pre-training to other types of mRNAs.
>
> 3. ### Over-reliance on single metric:
>
>     We reported Spearman correlation because it is the most widely used metric in RNA language modeling literature, adopted by recent prior works such as CodonBERT, SpliceBERT, and RNA-FM. However, we agree that additional metrics provide a more nuanced evaluation. To this end, we have additionally included Pearson correlation and R^2 as additional metrics in Appendix Table 10 (page 22 and highlighted in blue). These metrics confirm the same trend, with our HELM outperforming baselines across all reported metrics.
>
>     | Model (Pearson/$R^2$)   | AB1              | AB2               | MLOS               | Tc-Riboswitches   | mRFP              | COV-19 Vaccine    |
>     |-----------------------|------------------|-------------------|--------------------|-------------------|-------------------|-------------------|
>     | Transformer XE (MLM)  | 0.766 / 0.587    | 0.599 / 0.361     | 0.667 / 0.444      |  0.532 / 0.283    |  0.779 / 0.606    | 0.788 / 0.620     |
>     | Transformer HELM (MLM)| **0.793 / 0.629**| **0.603 / 0.364** |  **0.719 / 0.516** |  **0.601 / 0.361**| **0.847 / 0.717** |  **0.811 / 0.657**|
>
>     | Model (Pearson/$R^2$)   | AB1              | AB2               | MLOS               | Tc-Riboswitches   | mRFP              | COV-19 Vaccine    |
>     |-----------------------|------------------|-------------------|--------------------|-------------------|-------------------|-------------------|
>     | Transformer XE (CLM)  | 0.757 / 0.573    |  0.596/ 0.356     | **0.563 / 0.316**  |  0.485 / 0.235    | 0.880 / 0.774     |  0.762 / 0.580    |
>     | Transformer HELM (CLM)| **0.787 / 0.619**| **0.608 / 0.370** | 0.542 / 0.293      | **0.556 / 0.309** | **0.886 / 0.784** | **0.769 / 0.591** |
>
> [1] I Georgakopoulos-Soares et al. Secondary structures in RNA synthesis, splicing and translation. Comput Struct Biotechnol J. 2022.
>
> [2] B Alberts et al. The Generation of Antibody Diversity. Molecular Biology of the Cell. 4th edition. 2002.

---

> > ### Author Response · Authors · 2024-11-20
> > **Response to review and invitation for further discussion (Response 2 of 3)**
> >
> > 4. ### Performance analysis over various sequence lengths and GC content:
> >     We appreciate the valuable suggestion and have addressed the concerns regarding our evaluation methodology, particularly the analysis across sequence lengths and GC contents. To this end, we have expanded our evaluation to include a performance analysis of our HELM and XE baseline models, detailed in Appendix A.10 (pages 22-23 and highlighted in blue) of our paper. We utilized the iCodon mRNA thermostability prediction dataset, which contains sequences ranging from 30 to 3000 nucleotides in length. The GC content of these sequences varies broadly from 27.60% to 79.85%.
> >
> >     For the length analysis, we categorized the sequences into three length groups: 30-1000, 1000-2000, and 2000-3000 nucleotides, allowing us to analyze how model performance changes with increasing sequence length. For mRNA vaccines, the typical sequence length is between 1000-1500 nucleotides [3]. Therefore, our analysis of sequences in the 2000-3000 nucleotide range encompasses lengths that exceed those commonly found in mRNA vaccine datasets. For the GC content analysis, we categorized the dataset into three groups: GC ≤ 47%, 47% < GC ≤ 55%, and GC > 55%. Such grouping aligns with recognized thresholds in the literature, where GC content above ~56% is considered high ([4,5]).
> >
> >     Our findings indicate that the HELM models outperform the XE baseline across various conditions, highlighting the contribution of HELM’s hierarchical encoding mechanism.
> >
> >     ##### Sequence Length Partition
> >
> >     **CLM Results:**
> >
> >     | Length Range | XE     | HELM       |
> >     |--------------|--------|------------|
> >     | 0-1000       | 0.5232 | **0.5298** |
> >     | 1000-2000    | 0.4963 | **0.5278** |
> >     | 2000-3000    | 0.4675 | **0.5138** |
> >
> >     **MLM Results:**
> >
> >     | Length Range | XE     | HELM       |
> >     |--------------|--------|------------|
> >     | 0-1000       | 0.5200 | **0.5307** |
> >     | 1000-2000    | 0.5056 | **0.5289** |
> >     | 2000-3000    | 0.4747 | **0.5057** |
> >
> >     ##### GC Content Partition
> >
> >     **CLM Results:**
> >
> >     | GC Range  | XE     | HELM       |
> >     |-----------|--------|------------|
> >     | 0% - 47%  | **0.5676** | 0.5624 |
> >     | 47% - 55% | 0.4401 | **0.4846** |
> >     | 55% - 100%| 0.5167 | **0.5288** |
> >
> >     **MLM Results:**
> >
> >     | GC Range     | XE     | HELM       |
> >     |--------------|--------|------------|
> >     | 0% - 47%     | 0.5569 | **0.5700** |
> >     | 47% - 55%    | 0.4459 | **0.4958** |
> >     | 55% - 100%   | 0.5165 | **0.5237** |
> >
> >
> > 5. ### Biological interpretation of prediction errors:
> >     This biological relevance of prediction errors is the key inspiration for our HELM frameworks. As demonstrated in Fig. 1 (right), the prediction errors of the HELM pre-trained backbone align closely with a natural codon-to-amino acid mapping, assigning higher probabilities to synonymous codons. We would appreciate if the reviewer could further point us to any specific types of biological interpretation relevant to the presented tasks/datasets.
> >
> > 6. ### Biological justification of α parameter range selection:
> >     The α hyperparameter in the HXE loss controls the weighting between different hierarchical levels during HELM pre-training. Intuitively, setting α to 0 corresponds to a standard non-hierarchical method where each codon is treated independently. As α increases, the loss places greater emphasis on the hierarchical relationships, such as reduced penalization for prediction errors involving synonymous codons. Our empirical analysis in Appendix A.6, shows that moderate values of α (ranging from 0.2 to 0.4) consistently achieve optimal performance across various tasks as it allows to align model prediction errors with codon hierarchy while still being able to discriminate individual synonymous codons.
> >
> > [3] Gunter H.M. et al. mRNA vaccine quality analysis using RNA sequencing. Nat Commun 14. 2023.
> >
> > [4] Brown JC. High G+C Content of Herpes Simplex Virus DNA: Proposed Role in Protection Against Retrotransposon Insertion. Open Biochem J. 2007.
> >
> > [5] Courel et al. GC content shapes mRNA storage and decay in human cells. eLife. 2019.

---

> > > ### Author Response · Authors · 2024-11-20
> > > **Response to review and invitation for further discussion (Response 3 of 3)**
> > >
> > > 7. ### RNA vaccine design applications:
> > >     Our work directly addresses RNA vaccine design applications through the Ab1 and Ab2 expression datasets, where our HELM model demonstrates superior performance over existing baselines in both property prediction and generative sequence design (as shown in Table 1, 2, and Fig. 5 in the main text). We acknowledge that we had not emphasized enough the vaccine-design relevance of these datasets, and we have now clarified their importance in lines 245-248 (highlighted in blue). Also, we have expanded our evaluation with a new public dataset specifically designed for COVID-19 mRNA Vaccine Degradation Prediction [6] (lines 257-258 and highlighted in blue). Our results on this new dataset, presented in Tables 1 and 2 of the main text, further validate our hierarchical HELM model's effectiveness, as it consistently outperforms all baselines in this vaccine-related task. To facilitate the review, we draw attention to the comparative results for these datasets in the table below as well.
> > >
> > >    | Model                       | AB1 | AB2 | COV-19 Vaccine|
> > >    |-----------------------------|---------|---------|---------------|
> > >    |One-hot                      |0.431    |0.421    |    0.550      |
> > >    |RNA-FM                       |0.595    |0.515    |    0.742      |
> > >    |SpliceBERT                   |0.652    |0.542    |    0.757      |
> > >    |CodonBERT                    |0.686    |0.557    |    0.780      |
> > >    |Transformer XE (MLM) (Ours)  |0.748    |0.599    |    0.801      |
> > >    |Transformer XE (CLM) (Ours)  |0.752    |0.597    |    0.787      |
> > >    |Transformer HELM (MLM) (Ours)|**0.767**|*0.609*  |    **0.833**  |
> > >    |Transformer HELM (CLM) (Ours)|*0.760*  |**0.614**|    *0.789*    |
> > >
> > > 8. ### Narrow focus on antibody sequences and expression:
> > >    We respectfully disagree with the assessment. Firstly, we wish to clarify that in addition to antibody sequences, our work provides the results on five diverse non-antibody datasets (iCodon, Tc-Riboswitches, COVID-19 mRNA Vaccine, MLOS, mRFP) from multiple various species on both predictive and generative tasks in Tables 1, 2 and Fig.5. Secondly, we want to emphasize that antibody-focused datasets/tasks are themselves extremely relevant for therapeutics, with many recent ML papers at top conferences [7,8,9] specifically targeting the antibody domain due to its significant real-world impact.
> > >
> > > 9. ### Simplicity of codon bias analysis:
> > >
> > >     In its simplicity, our analysis effectively demonstrates the relationship between codon usage bias in the data and the predictive and generative performance improvements achieved by the HELM approach (Fig. 2). To further support codon usage analysis, we have now additionally included established Codon Pair Bias (CPB) [10] analysis that measures how frequently specific codon pairs occur in an organism's genome compared to their expected frequencies (Appendix A.5.2, lines 991-1004 on page 19, Appendix Fig. 7 on page 20 and highlighted in blue). The CPB analysis reveals the same trend with our HELM approach showing greater improvements on datasets with stronger codon bias. We have also referred to this analysis in main text lines 419-421 (highlighted in blue) now.
> > >
> > > 10. ### Performance on abnormal or mutated sequences/unusual codon usage patterns:
> > >     We appreciate this valuable suggestion regarding the evaluation on abnormal or mutated sequences. Such an analysis would require a dataset containing paired wild-type and mutated mRNA sequences with experimentally measured properties. To the best of our knowledge, no such datasets are currently publicly available. If the reviewer is aware of relevant datasets, we would greatly appreciate the opportunity to incorporate them into our analysis.
> > >
> > > [6] HK Wayment-Steele et al. Deep learning models for predicting RNA degradation via dual crowdsourcing. Nature Machine Intelligence. 2022.
> > >
> > > [7] Frey et al. Protein Discovery with Discrete Walk-Jump Sampling. ICLR. 2024.
> > >
> > > [8] Villegas-Morcillo et al. Guiding diffusion models for antibody sequence and structure co-design with developability properties. NeurIPS. 2023.
> > >
> > > [9] Zhu et al. Antibody Design Using a Score-based Diffusion Model Guided by Evolutionary, Physical and Geometric Constraints. ICML. 2024.
> > >
> > > [10] Roth et al. Measuring codon usage bias. Codon Evolution: Mechanisms and Models. 2015.

---

> > ### Comment · Reviewer_9ttH · 2024-11-25
> > **Continued question to the response 2**
> >
> > Thank you for mentioning the evolutionary process. Specifically, antibody data are derived from synthetic or screened datasets, where the optimization objectives of in vitro evolution are relatively straightforward and focused. In contrast, natural sequences evolve under multiple complex constraints and selective pressures in vitro conditions. Overall, the evolutionary processes of these two are fundamentally different. Could you please further clarify the mechanism of why antibody mRNA pre-training generalizes to other sequences? How to ensure the biological effectiveness of the model's generalization ability?

---

> ### Author Response · Authors · 2024-11-25
> **Requesting follow-up for rebuttal feedback and clarifications**
>
> We greatly appreciate the time and effort you’ve put into reviewing our submission. As the discussion period comes to a close tomorrow, we kindly request your feedback regarding our detailed responses to your valuable comments and the revisions we’ve made.
>
> Given our thorough responses and the comprehensive improvements made to the manuscript, we hope you will consider that our contribution merits a score well above marginal acceptance, reflecting its broader impact and technical depth. We believe we have effectively addressed all concerns, demonstrating the wide applicability and methodological novelty of our work.
>
> Your insights and updated evaluations would be incredibly helpful for the final assessment of our submission. Thank you once again for your thoughtful review and support during this process.

---

> ### Author Response · Authors · 2024-11-25
> **Response to the generalization question.**
>
> We agree that the generalization of antibody mRNA pre-training to natural RNA sequences is an interesting phenomenon and appreciate the reviewer's focus on this topic. As our experiments clearly demonstrate this generalization, we consider the following mechanisms may serve to explain it:
>
> 1. Conserved RNA Structural Features: RNA molecules, whether synthetic (e.g., antibody mRNA) or natural (e.g., viral mRNA, riboswitches), share conserved structural motifs such as loops, bulges, and stems that are critical for stability and function [1]. These commonalities likely enable the model to learn features that generalize across different RNA types.
>
> 2. Shared Molecular Interactions: Antibody mRNAs often encode features relevant to interacting with nucleic acid-like antigens, creating overlap in sequence characteristics with other RNA-based molecules, such as viral mRNA and riboswitches [2].
>
> We have now added this in Appendix A.12 (lines 1242-1256) and highlighted in blue. We would be happy to incorporate any additional suggestions to better highlight this phenomenon. We would also appreciate if the reviewer can clarify what is meant by biological effectiveness of model's generalization ability? And if the reviewer can clarify if all other raised issues are addressed in the rebuttal and if they help the reviewer consider a more positive evaluation of our work. Thanks!
>
> [1] Synthetic biology with RNA motifs, Saito & Inoue, Int J Biochem Cell Biol, 2009
>
> [2] The Generation of Antibody Diversity, Alberts et al., Molecular Biology of the Cell. 4th edition. 2002

---

> ### Author Response · Authors · 2024-12-01
> **Quantitative Analysis of mRNA Structural Motif Conservation Between Antibody and Non-Antibody mRNA Sequences to Quantitatively Address Reviewer's Question**
>
> We appreciate the opportunity to address this point regarding the generalizability of antibody (Ab) mRNA pre-training to other mRNA sequences. As detailed in our previous responses also, our hypothesis is not speculative but is rooted in established biological literature, which demonstrates conserved mRNA structural features and shared molecular interactions across antibody and non-antibody mRNA types. Specifically, previous studies [1, 2] (which we also cited in previous responses to the reviewer's question) have shown that both Ab and non-Ab mRNAs share similar structural motifs critical for stability and function, including conserved loops and bulges that contribute to secondary structure formation.
> To further substantiate this and respond to the reviewer’s query for biological validation of this, we have now conducted a detailed analysis that directly compares the mRNA secondary structures of one of our Ab mRNA datasets (Ab1) and four non-Ab mRNA datasets (Cov-19 Vaccine, Tc-Riboswitches, MLOS and mRFP) presented in the paper. For all these datasets, our model showed high predictive performance, making them ideal candidates for this analysis.
>
> ***Analysis Details***
> 1. Secondary Structure Prediction: We predicted RNA secondary structures for each sequence in the datasets using two widely used tools: EternaFold [3] and ViennaRNA [4]. The analysis considers structural motifs (e.g., loops and bulges), grouping contiguous unpaired bases together.
> 2. Motif-Based Similarity Metric: For each pair of sequences from the Ab and non-Ab mRNA datasets, we calculated a similarity score by comparing the size and position of unpaired motifs. A cumulative similarity score was generated by averaging pairwise scores across all comparisons, offering a single metric to quantify structural overlap between the datasets. The similarity scores range from 0 (no structural similarity) to 1 (perfect structural similarity).
> The similarity scores obtained using both prediction tools are summarized below:
>
> ### **Table 1: Motif Similarity Scores for Ab1 and Tc-Riboswitches dataset**
>
> | Metric                         | EternaFold Score | ViennaRNA Score |
> |--------------------------------|------------------|-----------------|
> | Mean Pairwise Motif Similarity | 0.40          | 0.45          |
>
> ---
>
> ### **Table 2: Motif Similarity Scores for Ab1 and Cov-19 Vaccine dataset**
>
> | Metric                         | EternaFold Score | ViennaRNA Score |
> |--------------------------------|------------------|-----------------|
> | Mean Pairwise Motif Similarity | 0.38            | 0.40           |
>
> ---
>
> ### **Table 3: Motif Similarity Scores for Ab1 and MLOS dataset**
>
> | Metric                         | EternaFold Score | ViennaRNA Score |
> |--------------------------------|------------------|-----------------|
> | Mean Pairwise Motif Similarity |      0.44       | 0.44          |
>
>
> ---
>
> ### **Table 4: Motif Similarity Scores for Ab1 and mRFP dataset**
>
> | Metric                         | EternaFold Score | ViennaRNA Score |
> |--------------------------------|------------------|-----------------|
> | Mean Pairwise Motif Similarity |      0.40       | 0.47           |
>
>
>
> Scores between 0.38 and 0.47, as observed in our study, indicate meaningful overlap in structural motifs, aligning with biological expectations for conserved secondary structure features across Ab and non-Ab mRNA types considered in our study, thus supporting the hypothesis of conserved mRNA structural features. The scores are consistent across two independent secondary structure prediction tools, reinforcing the robustness of the analysis. We will be happy to include this in our camera-ready version as well.
> While we believe this approach robustly addresses the reviewer’s query, we are still open to more specific suggestions from the reviewer for additional ways to further validate this.
>
> **Request for Score Consideration**
>
> We would also like to highlight that this analysis complements the extensive experimental validations and other investigations we already conducted to address the other questions of the reviewers. In light of the additional effort and clarity provided, we respectfully request that the reviewer considers the many experiments and results we provided for rebuttal in their evaluation and potentially increase their score to reflect the rigor and thoroughness of our proposed method and paper enhanced by valuable reviewer suggestions.
>
> [1] Synthetic biology with RNA motifs, Saito & Inoue, Int J Biochem Cell Biol, 2009
>
> [2] The Generation of Antibody Diversity, Alberts et al., Molecular Biology of the Cell. 4th edition. 2002
>
> [3] RNA secondary structure packages evaluated and improved by high-throughput experiments, Wayment-Steele, Nature Methods, 2022
>
> [4] ViennaRNA package 2.0., Lorenz et al.,  Algorithms for Molecular Biology, 6(1):26, 2011

---

> ### Author Response · Authors · 2024-12-02
> **Request for followup and score reconsideration based on rebuttal experiments and explanations**
>
> Dear Reviewer,
>
> As today marks the final day of the discussion period, we would like to express our gratitude for your thoughtful feedback. We have addressed all the questions raised in your review, as well as those from other reviewers, through additional experiments (where applicable) and detailed explanations during rebuttal, including analyses across multiple metrics, performance evaluations over different GC contents and sequence lengths, experiments using new mRNA vaccine data, and codon pair bias analyses to further substantiate the effectiveness of our proposed framework.
>
> Additionally, we have now provided computational analyses addressing your latest question regarding the biological effectiveness and generalization ability of our model (see our results in the previous thread). Based on the positive assessments of our work from other reviewers, as well as the rigorous experiments and comprehensive responses we have provided during rebuttal to raised questions and weaknesses, may we respectfully request if the reviewer can potentially consider increasing their score.
>
> Thank you for your valuable suggestions, which have greatly strengthened our work!

---

> ### Author Response · Authors · 2024-12-03
> **Request for rebuttal response and score reconsideration**
>
> Dear Reviewer,
>
> With only less than 2.5 hours remaining until the rebuttal deadline, we believe that we have successfully addressed your concerns and questions. If you find our responses satisfactory, we kindly request you to acknowledge our responses and consider raising your score. We truly appreciate your time, effort, and valuable contributions to improving our work.

---

### Official Review · Reviewer_TLr1 · 2024-11-05

**Soundness:** 4
**Presentation:** 4
**Contribution:** 4
**Rating:** 6
**Confidence:** 3

**Summary:**

The paper proposes an elegant and novel solution for incorporating codon hierarchical structure into language model training, which is hierarchical cross-entropy loss. Although the method is simple, the biological insight is deep and the performance is convincing. The authors also curate the hierarchical information and a pre-train dataset, which may benefit future research. Overall, I feel this would be an impactful work for the community.

**Strengths:**

1. The authors combine biological insight with language modeling.
2. The proposed HELM is solid and performs well for mRNA-related tasks. It achieves SOTA performance on several practical and impactful tasks, e.g., mRNA sequence design, and sequence region annotation.
3. The authors provide a curated dataset, and curated domain knowledge, which may benefit future computation-oriented research.
4. The authors provide solid benchmark experiments of various backbone architectures and tokenization methods.

**Weaknesses:**

1. The methodology itself is relatively simple.
2. It is not clear to what extent the model, the codes, and the data will be made public.

**Questions:**

I do have a strong concern for the reproducibility of this paper. Despite the fact that I think highly of this paper, I don't think the paper should be accepted without further declaration on reproducibility. Specifically, will the model checkpoint, source codes, and datasets be released?

---

> ### Author Response · Authors · 2024-11-20
> **Response to review (and invitation for further discussion)**
>
> We thank the reviewer for their positive feedback and appreciating the contributions of our work and the biological insights that our method enables. We address the valuable comments and questions raised by the reviewer point-by-point:
>
> 1. ### Declaration of reproducibility and if the model, code and data will be made public:
>      We appreciate the reviewer’s emphasis on reproducibility, which we take very seriously. Our group is fully committed to making all code, model weights, and data publicly available to ensure the complete reproducibility of all the results. In alignment with our organization's/institution’s policy, we are permitted to release code and data only after the paper’s acceptance. To address this, we have added a formal declaration of reproducibility at the end of the main text (page 11, highlighted in blue), where we explicitly commit to releasing all materials necessary to reproduce this work. Furthermore, we had also provided all model training and inference details in the Appendix Sections A.2 and A.3.
>
> 2. ### Simple methodology:
>      We want to emphasize that our work, to the best of our knowledge, is the first one to explicitly bridge known biological hierarchy and language models. With this work, we aim to demonstrate the significance of hierarchy, establish a hierachical baseline, and provide an experimental basis to foster future research in the direction of hierarchical bio-language models. From this perspective, we consider the technical simplicity of the proposed method as a strength. It enables seamless integration of biological hierarchy into any mRNA language model with minimal modifications while still considerably improving both predictive and generative aspects of a model.

---

> ### Author Response · Authors · 2024-11-25
> **Friendly request for feedback on responses to your questions...**
>
> We greatly appreciate the time and effort you’ve put into reviewing our submission. As the discussion period comes to a close tomorrow, we kindly request your feedback regarding our detailed responses to your valuable comments and the revisions we’ve made.
>
> Given our thorough responses and the comprehensive improvements made to the manuscript, we hope you will consider that our contribution merits a score well above marginal acceptance, reflecting its broader impact and technical depth. We believe we have effectively addressed all concerns, demonstrating the wide applicability and methodological novelty of our work.
>
> Your insights and updated evaluations would be incredibly helpful for the final assessment of our submission. Thank you once again for your thoughtful review and support during this process.

---

> ### Author Response · Authors · 2024-12-01
>
> As we approach the end of the discussion period, we kindly request your review of our detailed responses and revisions, which aims at comprehensively addressing all the concerns you raised. We respectfully request if would you be willing to increase your score if you are happy with how we have addressed your suggestions? Thank you!

---

### Author Response · Authors · 2024-11-25
**General Comment on Revisions and Additions**

We thank the reviewers for their valuable feedback and constructive suggestions, which have enhanced the quality and clarity of our submission. We are glad that the reviewers appreciate our elegant and novel solution  (***TLr1, 9ttH***) for incorporating codon hierarchy into mRNA language model, thus representing biologically-informed advancement in mRNA modeling (***eRzE***) while outperforming baselines by around 8% on various mRNA tasks (***TLr1, 9ttH, eRzE, SJYV***).

In terms of main contributions,
- We introduce HELM, a novel yet simple framework for mRNA language model pre-training, aligning learning with mRNA hierarchical structure (***TLr1, 9ttH***).
- We introduce a high-quality mRNA dataset (***9ttH, SJYV***) and evaluate diverse tokenization, pre-training, and LM architectures.
- HELM improves downstream performance by ~8% on diverse mRNA tasks, including property prediction, sequence generation, and region annotation (***TLr1, 9ttH, eRzE, SJYV***).

**Revisions during rebuttal**

Below, we summarize key changes made to the paper and additional experiments conducted in response to the reviewers’ comments:

## Declaration of Reproducibility
We have added a formal declaration of reproducibility in main text (page 11), explicitly committing to open-sourcing all code, model weights, and datasets after acceptance. Further, we provided implementation details for training and inference in Appendix Sec. A.2 and A.3 to ensure clarity and transparency.

## Inclusion of Additional Metrics
In addition to Spearman correlation, we now report Pearson correlation and R² values in Appendix Table 9. These metrics confirm the robustness of HELM’s performance across tasks and provide a more nuanced evaluation of the results.

## Performance Analysis on Sequence Lengths and GC Content
We conducted detailed experiments to evaluate HELM’s performance across varying sequence lengths and GC content ranges, utilizing the iCodon mRNA thermostability prediction dataset. The results, presented in Appendix A.10, demonstrate that HELM consistently outperforms baselines across these diverse tasks.

## Scaling Experiments
To explore scalability, we conducted additional experiments with 100M parameter models. The results, included in Appendix A.7, reveal that HELM’s hierarchical pre-training consistently outperforms non-hierarchical baselines even at larger scales, reinforcing utility of explicitly incorporating hierarchical priors.

## Codon Pair Bias (CPB) Analysis
We have now included an established Codon Pair Bias (CPB) analysis, which measures frequency of specific codon pairs in an organism's genome compared to their expected frequencies. This analysis is detailed in Appendix A.5.2 (lines 991-1004 on page 19) and visually presented in Appendix Fig. 7 (page 20, highlighted in blue). This analysis reveals that HELM shows greater improvements on datasets with stronger codon bias. We have also referred to this analysis in the main text (lines 419-421, highlighted in blue).

## Alternative Baseline Experiments
We implemented a reviewer-suggested baseline by introducing separate embeddings for codon types combined with token embeddings. The results, included in Appendix A.11, show that HELM still outperforms this alternative baseline across all datasets, underscoring the advantages of hierarchical pretraining.

## New Dataset and Expanded Vaccine Relevance
We incorporated a new public COVID-19 mRNA Vaccine Degradation Prediction dataset to further evaluate HELM’s relevance for RNA vaccine design. Results on this dataset, presented in Tables 1 and 2, demonstrate HELM’s effectiveness in this critical application area.

## Improved Biological Justifications
We have expanded the discussion on HELM’s biological relevance, including:
- Generalizability to diverse mRNA types due to shared structural and functional motifs.
- Alignment of prediction errors with synonymous codons for better biological interpretability.
- The role of the α parameter in the hierarchical loss, allowing flexible weighting of hierarchical levels.

## Clarification on Data Splitting
To prevent data leakage, we detailed clustering-based partitioning methodologies (e.g., LinClust with a similarity threshold of 0.9) for datasets without predefined splits. These details are now clarified in the main text.

## Quantitative Analysis of mRNA Structural Motif Conservation Between Antibody and Non-Antibody mRNA Sequences to Demonstrate Generalizability of Pre-Trained Models
We conducted a quantitative analysis of RNA secondary structure conservation between antibody and non-antibody mRNA datasets using motif-based similarity metrics, demonstrating meaningful structural overlap that supports the generalizability of our model’s learned features across diverse mRNA types.

These revisions and additions address the reviewers’ comments comprehensively, strengthening the contributions of our work. We thank the reviewers for their insights, which have improved our submission.

---

### Meta-Review · Area_Chair_jVm7 · 2025-01-03

**Metareview:**

The paper presents a pretraining strategy that incorporates codon-level hierarchical structure into language model training. The reviewers found the paper's methodology to be a useful combination of biological insights with language modeling. While the technical novelty from the ML point of view is limited, the method's significance to bioinformatics and the solid experimental results make up for this concern. There were some concerns about reproducibility and the evaluation, but these were addressed during the rebuttal period. Given all this, I am happy to recommend acceptance. Please incorporate all the reviewer comments in the final version, and make sure to release the model, code, and data.

**Additional Comments On Reviewer Discussion:**

The authors diligently posted numerous rebuttal comments. Some, but not all, reviewers responded to these comments.

---

### Decision · Program_Chairs · 2025-01-22

Accept (Poster)